# InertialAR: Autoregressive 3D Molecule Generation with Inertial Frames

## Abstract

Transformer-based autoregressive models have emerged as a unifying paradigm across modalities such as text and images, but their extension to 3D molecule generation remains underexplored. The gap stems from two fundamental challenges: (1) tokenizing molecules into a canonical 1D sequence of tokens that is invariant to both SE(3) transformations and atom index permutations, and (2) designing an architecture capable of modeling hybrid atom-based tokens that couple discrete atom types with continuous 3D coordinates. To address these challenges, we introduce InertialAR. InertialAR devises a canonical tokenization that aligns molecules to their canonical inertial frames and reorders atoms to ensure SE(3) and permutation invariance. Moreover, InertialAR equips the attention mechanism with geometric awareness via geometric rotary positional encoding (GeoRoPE). In addition, it utilizes a hierarchical autoregressive paradigm to predict the next atom-based token, predicting the atom type first and then its 3D coordinates via Diffusion loss. Experimentally, InertialAR achieves state-of-the-art performance on 7 of the 10 evaluation metrics for unconditional molecule generation across QM9, GEOM-Drugs, and B3LYP. Moreover, it significantly outperforms strong baselines in controllable generation for targeted chemical functionality, attaining state-of-the-art results across all 5 metrics.

## 1 Introduction

Autoregressive (AR) models have achieved substantial progress in artificial intelligence (AI) in recent years. In natural language processing, their strong sequence modeling capability and scalability have established them as the de facto architecture for foundation models (Brown et al., 2020; Touvron et al., 2023; Achiam et al., 2023). Moreover, they have shown competitive performance on par with diffusion models in image generation, suggesting their viability as a unified sequence modeling paradigm (Sun et al., 2024; Tian et al., 2024). Inspired by their success across these diverse modalities, we seek to investigate whether AR models can serve as an effective generative model paradigm for 3D molecule generation.

While diffusion models have achieved impressive results in 3D molecule generation, they are often limited by computationally intensive iterative sampling and a lack of flexibility for variable-length generation (Hoogeboom et al., 2022; Xu et al., 2023; Vignac et al., 2023). In contrast, AR models offer a compelling alternative: by casting 3D molecule generation as a sequence prediction problem, they enable highly efficient and flexible generation of variable-sized molecules.

However, adapting AR models for 3D molecule generation poses unique challenges at both data and model levels. On the data side, the key difficulty centers on tokenizing 3D molecules into 1D sequences of tokens compatible with Transformer-like models. An ideal tokenization must satisfy two criteria: (1) SE(3)-equivariance, *i.e.*, equivariant tokenization under rotations and translations, and (2) permutation invariance of the atom indexing to establish a canonical sequence order for each molecule. On the model side, unlike conventional AR models that merely predict the next discrete token at each step, the AR model for 3D molecule generation requires jointly predicting a discrete atom type (*e.g.*, C, H, O, N) and its continuous 3D coordinates, due to the dual chemical and geometric information encoded in each atom.

**Our Contributions.** To address these challenges, we propose InertialAR, a novel AR model for 3D molecule generation. InertialAR rests on two key innovations. First, it leverages a canonical tok-

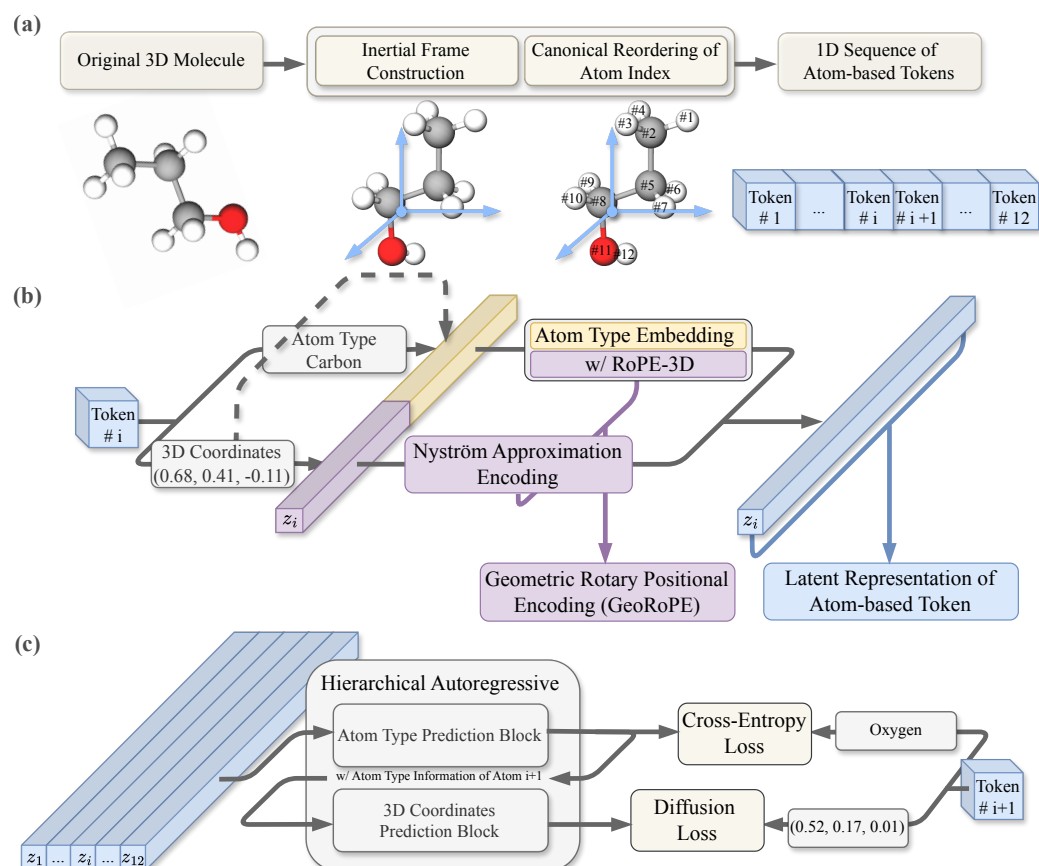

Figure 1: Overview of InertialAR: (a) canonical tokenization, (b) geometric rotary positional encoding (GeoRoPE), and (c) hierarchical autoregressive paradigm.

enization strategy that uses a canonical inertial frame to align 3D molecules, converting them into a sequence of atom-level tokens that ensures SE(3) equivariance. It subsequently applies a canonical reordering of its atoms to guarantee atom index permutation invariance. Second, it introduces Geometric Rotary Positional Encoding (GeoRoPE), which injects relative positional awareness and pairwise distance information between atoms into the attention mechanism, making it geometry-aware. Built upon these two components, InertialAR employs a hierarchical AR paradigm for density estimation, iteratively predicting the discrete atom types using cross-entropy and continuous atom positions using diffusion loss.

To evaluate the effectiveness of InertialAR, we conduct comprehensive experiments on both unconditional and controllable generation. For unconditional generation, InertialAR achieves state-of-the-art results on 3 of 6 key metrics on QM9 and GEOM-Drugs. To further assess its scalability and robustness, we evaluate on the more challenging large-scale B3LYP dataset, where InertialAR attains state-of-the-art performance across all 4 metrics, clearly surpassing other prominent diffusion and AR models. Furthermore, on the more demanding task of class-conditional generation, InertialAR combined with classifier-free guidance establishes state-of-the-art results on all 5 evaluation metrics, enabling targeted generation and editing of molecules with desired chemical functionality.

**Related work.** We briefly review the most related works here and include a more detailed overview in Appendix B. The central requirement for 3D molecule generation is respecting SE(3) symmetry. Existing methods can be grouped into four paradigms: (i) SE(3)-equivariant architectures (Thomas et al., 2018a; Liao & Smidt, 2023; Schütt et al., 2021; Satorras et al., 2022b), (ii) invariant-feature modeling (Schütt et al., 2017; Gasteiger et al., 2022), (iii) data augmentation (Flam-Shepherd & Aspuru-Guzik, 2023; Abramson et al., 2024), and (iv) input canonicalization (Antunes et al., 2024; Yan et al., 2024; Li et al., 2024b; Fu et al., 2024). Another key challenge for autoregressive 3D

generation is tokenization. While recent studies have investigated text sequence-based tokenization (Li et al., 2024b; Yan et al., 2024; Flam-Shepherd & Aspuru-Guzik, 2023), some parallel works are concurrently exploring voxel-based approaches (Faltings et al., 2025; Lu et al., 2025b). However, they both rely on spatial discretization, which discards fine-grained geometry and fails to preserve atom-level granularity.

## 2 PRELIMINARIES

**3D Molecule Generation.** The goal of 3D molecule generation is to directly construct physically plausible 3D molecular conformations. Formally, a 3D molecule with $n$ atoms can be represented as a point cloud $\mathcal{M} = (t, C)$. The vector $t = [t_1, \cdots, t_n] \in \mathbb{Z}^n$ encodes the atom types, where $t_i$ denotes the nuclear charge of the $i$-th atom. The coordinate matrix $C = [c_1, \cdots, c_n] \in \mathbb{R}^{3 \times n}$ specifies the 3D position of each atom, with $c_i \in \mathbb{R}^3$.

**Autoregressive Models and Tokenization of 3D Molecule.** Autoregressive (AR) models address sequence modeling by framing it as a "next-token prediction" problem. This approach, a direct application of the chain rule of probability, factorizes the joint distribution of a sequence $x = (x_1, \ldots, x_n)$ into a product of conditional probabilities:

$$p(x) = p(x_1, \ldots, x_n) = \prod_{i=1}^{n} p(x_i | x_1, \ldots, x_{i-1}). \tag{1}$$

The model's core task is thus to learn the conditional distribution $p(x_i|x_{<i})$ for each step, which is typically parameterized by a powerful neural network such as the Transformer (Vaswani, 2017). The primary challenge in applying AR models to 3D molecular generation lies in the effective tokenization of a 3D molecular structure into a 1D sequence of tokens suitable for Transformer architectures.

**Class-conditional Generation and Classifier-free Guidance.** Class-conditional generation produces samples conditioned on a class label $c$ (Esser et al., 2021; Peebles & Xie, 2023). Classifier-free guidance (CFG), originally proposed by Ho & Salimans (2022), enhances both sample quality and conditional alignment. It trains a single model on both the conditional distribution $p(x|c)$ and the unconditional distribution $p(x)$ by randomly dropping labels during training. Then during inference, conditional generation is steered by combining the two predictions:

$$p_g = p_u + s(p_c - p_u), \tag{2}$$

where $p_c$ and $p_u$ denote the conditional and unconditional predictions, respectively, and $s$ is the guidance scale controlling the trade-off between class fidelity and sample diversity.

## 3 INERTIALAR

The Inertial Autoregressive Model (InertialAR) casts 3D molecule generation as an AR process, where a molecule is sequentially built by predicting "the next atom-based token" at each step. To achieve this, a 3D molecule $\mathcal{M}$ is tokenized into an ordered 1D sequence of $n$ atom-based tokens, $\mathcal{M} = (a_1, \ldots, a_n)$, where each atom-based token $a_i = (t_i, c_i)$ contains a discrete atom type $t_i$ and continuous 3D coordinates $c_i = (x_i, y_i, z_i)$. Thus, the corresponding probability factorizes as:

$$p(\mathcal{M}) = \prod_{i=1}^{n} p(a_i | a_{<i}) = \prod_{i=1}^{n} p((t_i, c_i) | a_{<i}). \tag{3}$$

### 3.1 CANONICAL TOKENIZATION OF 3D MOLECULES

The factorization in Equation (3) makes AR models inherently sensitive to the token order. Therefore, a robust tokenization must be invariant to two fundamental symmetries: the continuous SE(3)-equivariance of the molecular geometry under rotations and translations, and the discrete permutation symmetry of the atom indexing (which can yield up to $n!$ permutations for $n$ atoms). Such a canonical tokenization ensures that each molecule maps to a unique token sequence, eliminating ambiguity and enabling effective learning.

More concretely, we introduce a two-step canonical tokenization, as shown in Figure 1(a). First, to address SE(3) symmetry, we align the molecular system to its canonical inertial frame, resulting

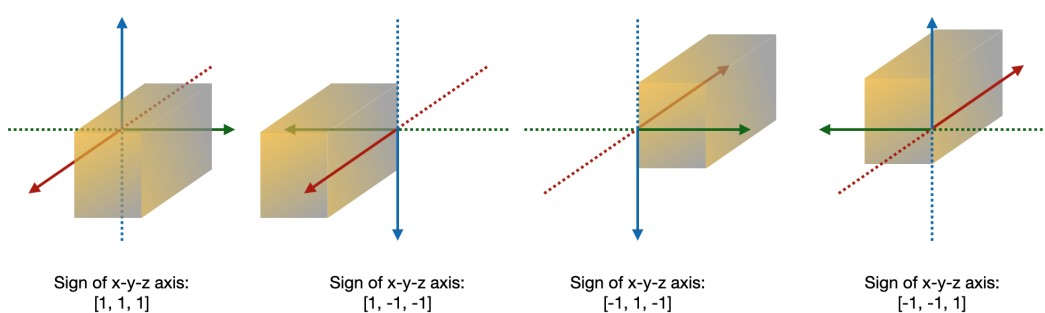

Figure 2: Illustration of introducing a fourth node as the anchor node. We define the sign of the x-y-z axis to make sure that $x_4$ is in the first quadrant, and there are four cases as illustrated in the four subfigures.

in an invariant canonical pose. Second, to address index permutation symmetry, the atoms are deterministically reordered according to a predefined rule. More details are explained below.

**Step 1: Canonical Inertial Frame Construction**. First, we employ the following four steps to derive the reference frames that construct the rotation matrix from $N$ atomic positions $c$: (1) Calculate the mass center: $\bar{c} = \frac{1}{N} \sum_i c_i$. (2) Adjust position relative to the center $c_i = c_i - \bar{c}$. (3) Compute the inertia tensor $\hat{I} = \sum_i \|c_i\|^2 I - c_i c_i^T$, where $I$ is the unit diagonal matrix.

**How to define the orderings of canonical inertial frame axes?** We follow the ordering of the eigenvalues to define the ordering of the eigen-vectors, which form the rotation matrix. The key point to note is how to handle the tie between the eigenvalues. In such cases, the molecular system is symmetric (*e.g.*, $CO_2$ or $CH_4$), leading to degenerate eigenvalues of the inertia tensor. Consequently, the canonical inertial frame is not uniquely defined, yet all valid frames are physically equivalent.

**How to define the directions of canonical inertial frame axes?** The orthonormal $I$ is the basis. Meanwhile, there are eight possible combinations for the directions or signs of the x-, y-, and z-axes, given by $\{\pm 1, \pm 1, \pm 1\}$, respectively. First, we enforce the ordering of the x-y-z axis to be right-handed, *i.e.*, the determinant of $I$ to be 1, not -1. This still gives us four possible combinations. Then we can define a unique direction for each molecule system by introducing a fourth node, as in Theorem 1.

**Theorem 1.** *For an inertial frame $F$, we build up the corresponding right-handed axes as coordinate systems $Q$. Then we need to incorporate a fourth point that is not on the y-z plane or x-z plane to uniquely determine the directions of the coordinate system with one rotation transformation matrix.*

For detailed proof, please check Appendix E. As illustrated in Theorem 1, we must include a fourth node to uniquely determine the directions of the three axes. To achieve this, we consider a fourth node $x_4$ that is not on the y-z plane or x-z plane and has the largest distance to the origin. Then we define the *requirement* that x-$x_4$-z and $x_4$-y-z are also right-handed; in other words, this requirement is essentially saying that $x_4$ should be in the first quadrant of the x-y plane. For implementation, $x_4$ is a 3D point whose projection onto the x–y plane falls into one of the four quadrants: the first, second, third, or fourth quadrant, depending on the signs of its x and y coordinates. Each of them defines the signs (or directions) of the canonical inertial frame axes, as illustrated in Figure 2.

**Step 2: Canonical Reordering of Atom Index**. To resolve the discrete permutation ambiguity of atom indexing, we first process the 3D molecular structure with RDKit (Landrum, 2016) to obtain a molecule object, which provides the corresponding attributed molecular graph with atoms as nodes and bonds as edges. Each atom is first assigned an initial identifier based on intrinsic chemical and topological features (*e.g.*, atomic number, degree, formal charge, attached hydrogens, ring membership). These identifiers are then iteratively refined by aggregating information from neighboring atoms until stabilization. Atoms are finally ordered according to their refined identifiers, ensuring that isomorphic molecules map to the same sequence. For cases where symmetry leaves multiple atoms indistinguishable, a deterministic tie-breaking procedure perturbs the identifiers and re-runs refinement until a unique order is obtained (Landrum, 2016). Such canonical reordering reduces the $n!$ possible permutations to a unique ordering, providing the consistent input required for AR learning.

## 3.2 GeoRoPE: Geometric Rotary Positional Encoding

After obtaining the canonical sequence of tokens, each atom-based token $a_i = (t_i, c_i)$ defined in Equation (3) must be effectively encoded into a latent representation suitable for Transformer modeling. This representation should capture both the discrete atom type $t_i$ and the continuous 3D coordinates $c_i = (x_i, y_i, z_i)$, ensuring that the self-attention mechanism can fully perceive and reason about the chemical identity and spatial arrangement of atoms.

**Atom Type Embedding**. For the discrete atom type $t_i$, we employ a learnable embedding layer to map this categorical feature into a continuous, high-dimensional vector:

$$z_i^{\text{type}} = \text{Embedding}(t_i). \tag{4}$$

**Geometric Rotary Positional Encoding (GeoRoPE).** To enable the self-attention mechanism to effectively capture the relative spatial relationships between atoms, a geometry-aware encoding of the continuous 3D coordinates $c_i = (x_i, y_i, z_i)$ is essential. To this end, we introduce GeoRoPE, the Geometric Positional Encoding tailored for 3D point-based tokens, as shown in Figure 1(b). GeoRoPE integrates: (i) **3D Rotary Positional Encoding (RoPE-3D)** for relative positional awareness along spatial axes, and (ii) **Nyström Approximation Encoding** for efficient modeling of pairwise distances.

**(i) 3D Rotary Positional Encoding for Continuous 3D Coordinates.** To make the self-attention mechanism geometry-aware, the positional encoding must ensure the inner product for absolute positions $c_i$ and $c_j$ depends solely on their relative positions, $c_j - c_i$. This can be expressed as:

$$R_{c_i}^T R_{c_j} = R_{x_i,y_i,z_i}^T R_{x_j,y_j,z_j} = R_{x_j-x_i,y_j-y_i,z_j-z_i} = R_{c_j-c_i}. \tag{5}$$

Here, $R_{x,y,z}$ is the positional encoding function that maps 3D coordinates to their high-dimensional representation. This forces the attention scores to reflect the molecule's internal geometry, not its arbitrary global orientation. Then, inspired by Su (2021), we propose the 3D Rotary Positional Encoding (RoPE-3D) for atom-based tokens in the Euclidean space:

$$R_{x,y,z}\boldsymbol{q} = \begin{bmatrix} q_0 \\ q_1 \\ q_2 \\ q_3 \\ q_4 \\ q_5 \end{bmatrix} \cdot \begin{bmatrix} \cos x\theta_0 \\ \cos x\theta_0 \\ \cos y\theta_0 \\ \cos y\theta_0 \\ \cos z\theta_0 \\ \cos z\theta_0 \end{bmatrix} + \begin{bmatrix} -q_1 \\ q_0 \\ -q_3 \\ q_2 \\ -q_5 \\ q_4 \end{bmatrix} \cdot \begin{bmatrix} \sin x\theta_0 \\ \sin x\theta_0 \\ \sin y\theta_0 \\ \sin y\theta_0 \\ \sin z\theta_0 \\ \sin z\theta_0 \end{bmatrix}. \tag{6}$$

This RoPE-3D in Equation (6) is then applied to the query $\boldsymbol{q}$ and key $\boldsymbol{k}$ vectors of each atom within the self-attention mechanism. A crucial outcome of this formulation is that the inner product between a query vector transformed by position $c_i$ and a key vector transformed by position $c_j$ becomes a function of only their relative positions, $c_j - c_i$:

$$(R_{c_i}\boldsymbol{q})^T (R_{c_j}\boldsymbol{k}) = (R_{x_i,y_i,z_i}\boldsymbol{q})^T (R_{x_j,y_j,z_j}\boldsymbol{k}) = \boldsymbol{q}^T R_{x_j-x_i,y_j-y_i,z_j-z_i}\boldsymbol{k} = \boldsymbol{q}^T R_{c_j-c_i}\boldsymbol{k}. \tag{7}$$

Consequently, the attention score between any two atoms depends on their feature representations (via $\boldsymbol{q}$ and $\boldsymbol{k}$) and their relative spatial arrangement, fulfilling the initial requirement for a geometry-aware self-attention mechanism.

**(ii) Nyström Approximation Encoding for Pairwise Distance.** One limitation of using RoPE-3D in Equation (6) for structure tokenization is that it treats each axis separately; though by expectation, it should be able to learn the token pairwise distance information. We empirically observe that merely using RoPE-3D cannot learn adequate information, while explicitly adding the pairwise information is more informative.

Then the question is how to explicitly incorporate the pairwise distance into the model. One straightforward way is to directly inject the distance information into the attention score, like (Shi et al., 2023). However, such an architecture is not compatible with the standard transformer architecture used in large language models (Bai et al., 2023; Achiam et al., 2023; Touvron et al., 2023).

To alleviate this issue, we consider the Nyström method (Williams & Seeger, 2000). It is a low-rank approximation to obtain the pairwise distance. More concretely, suppose we have a Gram matrix over $n$ points, *i.e.*, $K \in \mathbb{R}^{n \times n}$. Each element $K_{ij}$ is the radial basis function (RBF) over the distance between $i$-th and $j$-th points, $K_{ij} = RBF(c_i, c_j) = \exp(-\frac{\|c_i-c_j\|^2}{2\sigma^2})$, with $c_i$ denoting

the 3D coordinates of the $i$-th point in an Euclidean space. Then we sample $m$ anchor points, $(c_1, c_2, ..., c_m)$ with $m \ll n$. The RBF of these $m$ points can compose an $m$-rank matrix $A \in \mathbb{R}^{m \times m}$ with positive eigenvalues. By Cholesky decomposition, we have $A = LL^T$. Then, to obtain the RBF of a new point pair $K(i, j)$, we first construct the feature between point $i, j$ and the $m$ anchor points as $k_i = [\text{RBF}(i, 0), \text{RBF}(i, 1), ..., \text{RBF}(i, m)]^T \in \mathbb{R}^{m \times 1}$. For each atom $i$, we define its Nyström approximation encoding as:

$$z_i^{\text{Nyström}} = L^{-1} k_i. \tag{8}$$

This allows the approximated RBF, which encodes the pairwise distance information between atoms, to be recovered directly by the inner product in the attention mechanism (details are in Appendix D):

$$\tilde{k}(i, j) = (z_i^{\text{Nyström}})^T (z_j^{\text{Nyström}}). \tag{9}$$

**Latent Representation of Atom-based Token**. The final input representation for each atom $i$ is the concatenation of its type embedding and its Nyström approximation encoding:

$$z_i = [z_i^{\text{type}}, z_i^{\text{Nyström}}]. \tag{10}$$

Within the attention layer, the input representation $z_i$ is projected into query $q_i$, key $k_i$, and value $v_i$. Here, we take the query projection for illustration:

$$q_i = W_q z_i. \tag{11}$$

Crucially, to maintain the distinct roles of the atom type embedding and Nyström approximation encoding, the weight matrix $W_q$ is structured as a block-diagonal matrix. This structure ensures that the two components of the input representation are projected independently. Recall that $z_i = [z_i^{\text{type}}, z_i^{\text{Nyström}}]$, the projection is implemented as:

$$\begin{bmatrix} q_i^{\text{type}} \\ q_i^{\text{Nyström}} \end{bmatrix} = \begin{bmatrix} W_q^{\text{type}} & 0 \\ 0 & W_q^{\text{Nyström}} \end{bmatrix} \begin{bmatrix} z_i^{\text{type}} \\ z_i^{\text{Nyström}} \end{bmatrix}, \tag{12}$$

where $W_q^{\text{type}}$ is the learnable weight matrix for the type component, and $W_q^{\text{Nyström}}$ is the identity matrix. The key $k_i$ and value $v_i$ are computed in an analogous manner using their own block-diagonal weight matrices, $W_k$ and $W_v$. The 3D Rotary Positional Encoding is applied only to the atom type components. The final query $\tilde{q}_i$ and key $\tilde{k}_j$ vectors used in the attention score calculation are then formed by concatenating these two parts:

$$\tilde{q}_i = \begin{bmatrix} R_{c_i} q_i^{\text{type}} \\ q_i^{\text{Nyström}} \end{bmatrix}, \quad \tilde{k}_j = \begin{bmatrix} R_{c_j} k_j^{\text{type}} \\ k_j^{\text{Nyström}} \end{bmatrix} \tag{13}$$

The key advantage of this construction is revealed in the inner product, which combines the two sources of geometric information. The attention score between atoms $i$ and $j$ is computed as:

$$\begin{aligned} \text{AttentionScore}(i, j) &= \tilde{q}_i^T \tilde{k}_j \\ &= (R_{c_i} q_i)^T (R_{c_j} k_j) + (q_i^{\text{Nyström}})^T (k_j^{\text{Nyström}}) \\ &= \underbrace{q_i^T R_{c_j - c_i} k_j}_{\text{Relative Position from RoPE-3D}} + \underbrace{\text{RBF}(\|c_i - c_j\|)}_{\text{Pairwise Distance from Nyström}} \end{aligned} \tag{14}$$

This formulation ensures that the self-attention score explicitly and simultaneously models both the relative geometric arrangement via RoPE-3D and the pairwise distance via the Nyström approximation encoding, providing a rich and robust inductive bias.

### 3.3 Hierarchical Autoregressive Architecture

The sequence of latent representations derived from Section 3.2, $(z_1, \ldots, z_n)$, is then processed by the autoregressive Transformer backbone to produce a sequence of context-aware hidden embeddings, $(h_1, \ldots, h_n)$. At each step $i$, the hidden embedding $h_i$, which encapsulates the full context of the previous atoms $a_{<i+1}$, is used to predict the next token, $a_{i+1} = (t_{i+1}, c_{i+1})$. This presents a unique challenge, as the prediction target is a hybrid of a discrete type and a continuous coordinate vector. To address this, we factorize the conditional probability into two components:

$$p(a_{i+1} \mid h_i) = p(a_{i+1} \mid h_i) = p(t_{i+1} \mid h_i) \cdot p(c_{i+1} \mid t_{i+1}, h_i). \tag{15}$$

In Equation (15), the model first predicts the atom type $t_{i+1}$ conditioned on the hidden embedding $h_i$. Subsequently, the continuous 3D coordinates $c_{i+1}$ are predicted given both $t_{i+1}$ and $h_i$.

Concretely, we implement this using a hierarchical AR architecture (as illustrated in Figure 1(c)): (i) a type-prediction block dedicated to modeling the discrete, categorical distribution over atom types, and (ii) a coordinates-prediction block to predict continuous 3D coordinates. This hierarchical architecture not only aligns with the intrinsic nature of molecular generation but also enhances learning efficiency by decoupling the tasks of categorical classification and continuous density estimation (Cheng et al., 2025b).

**Cross-Entropy Loss for Type Prediction Block.** For the discrete atom type $t_{i+1}$, we employ the standard cross-entropy, which directly maximizes the likelihood of the ground-truth atom type given the hidden embedding $h_i$:

$$\mathcal{L}_{\text{type}} = -\mathbb{E}_{(h_i, t_{i+1}) \sim \mathcal{D}} \left[ \log p_\theta(t_{i+1} \mid h_i) \right]. \tag{16}$$

**Diffusion Loss for 3D Coordinates Prediction Block.** Autoregressive models are naturally well-suited for generating discrete tokens using cross-entropy. However, for continuous 3D coordinates $c_{i+1}$, we empirically find that direct regression yields poor performance. To overcome this limitation, we adopt Diffusion Loss from Li et al. (2024a), which provides an effective framework for extending autoregressive models to continuous-valued token generation. The high-level idea is that we perturb the ground-truth position $c_{i+1}$ by adding Gaussian noise with a sampled noise level $\sigma$, and train a denoising network $\epsilon_\theta$ to recover the injected noise (Karras et al., 2022). Concretely, the perturbed coordinate is given by

$$c_{i+1}^{(\sigma)} = c_{i+1} + \sigma\,\epsilon, \quad \epsilon \sim \mathcal{N}(0, I). \tag{17}$$

Conditioned on the hidden embedding $h_i$ and the predicted atom type $t_{i+1}$, the denoising network is optimized with the following loss function:

$$\mathcal{L}_{\text{diff}} = \mathbb{E}_{\sigma, c_{i+1}, \epsilon} \left[ \left\| \epsilon - \epsilon_\theta(c_{i+1}^{(\sigma)}, \sigma, t_{i+1}, h_i) \right\|_2^2 \right]. \tag{18}$$

This objective enables the coordinates prediction block to model the continuous distribution of atom positions. At inference time, atom coordinates are generated by iterative denoising from Gaussian noise, conditioned on both the autoregressive context $h_i$ and the sampled atom type $t_{i+1}$.

**Controllable Generation with Classifier-free Guidance.** We incorporate classifier-free guidance (CFG, details in Section 2) into InertialAR to enable controllable generation. During inference, CFG modulates conditional generation process by leveraging the difference between conditional and unconditional predictions:

$$p_g = p_u + s(p_c - p_u), \tag{19}$$

where $p_c$ and $p_u$ denote the conditional and unconditional predictions, respectively, and $s$ is the guidance scale. In InertialAR, CFG is applied to the estimated noise $\epsilon_\theta$ in diffusion for coordinates generation, as well as to the logits over a discrete vocabulary for atom type prediction. By tuning $s$, we can achieve both stronger adherence to target molecular classes and better structural validity.

## 4 EXPERIMENTS

### 4.1 UNCONDITIONAL 3D MOLECULE GENERATION

**QM9 and GEOM-Drugs Dataset.** We use QM9 (Ramakrishnan et al., 2014) and GEOM-Drugs (Axelrod & Gómez-Bombarelli, 2022) for unconditional 3D molecular generation. QM9 contains 130K small molecules with high-quality 3D conformations (up to 9 heavy atoms). We split the dataset into train, validation and test sets with 100K, 17K and 13K samples, separately. GEOM-Drugs consists of 37M conformations for around 450K unique molecules (up to 181 atoms and 44.2 atoms on average). Following Hoogeboom et al. (2022), we select the 30 lowest-energy conformations per molecule for training. **B3LYP Dataset.** Moreover, we evaluate on a brand new, larger, and more comprehensive 3D molecular dataset, the PubChemQC B3LYP/6-31G//PM6 dataset (abbreviated as B3LYP) (Nakata & Maeda, 2023). This dataset contains a total of 85,938,443 molecules, covering a wide range of chemical diversity with molecular weights up to 1000 and more than 50 different atom types. We use a subset of 1M molecules for training. The evaluation metrics remain consistent with those used for the QM9 and GEOM-Drugs datasets.

**Evaluation.** Model performance is assessed through a set of chemical feasibility metrics. Bond types (single, double, triple, or none) are determined from molecular geometries based on pairwise atomic distances and atom identities. The evaluation includes Atom Stability (proportion of

Table 1: Unconditional generation on 3D molecules on QM9 and GEOM-Drugs.

| Methods | QM9 | | | | GEOM-Drugs | |
|---|---|---|---|---|---|---|
| | Valid (%) | Valid&Uni (%) | AtomSta (%) | MolSta (%) | Valid (%) | AtomSta (%) |
| EFN | 40.2 | 39.4 | 85.0 | 4.9 | – | – |
| G-SchNet | 85.5 | 80.3 | 95.7 | 68.1 | – | – |
| GDM | – | – | 97.0 | 63.2 | 90.8 | 75.0 |
| GDM-AUG | 90.4 | 89.5 | 97.6 | 71.6 | 91.8 | 77.7 |
| EDM | 91.9 | 90.7 | 98.7 | 82.0 | 92.6 | 81.3 |
| MiDi | **97.9** | **97.0** | 97.9 | 84.0 | 78.0 | 82.2 |
| GeoLDM | 93.8 | 92.7 | 98.9 | 89.4 | **99.3** | 84.4 |
| UniGEM | 95.0 | 93.2 | 99.0 | 89.8 | 98.4 | 85.1 |
| Geo2Seq | 97.1 | 81.7 | 98.9 | 93.2 | 96.1 | 82.5 |
| InertialAR | 97.4 | 92.5 | **99.3** | **94.7** | 96.8 | **87.2** |

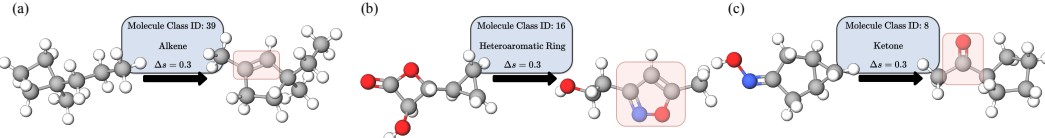

Figure 3: Visualization of molecule editing by tuning the CFG guidance scale $s$.

atoms satisfying correct valency), Molecule Stability (proportion of molecules in which all atoms are stable), Validity (fraction of chemically valid molecules as verified by RDKit), and Uniqueness (fraction of non-duplicate molecules among generated samples). All metrics are computed following evaluation protocols established in prior work (Hoogeboom et al., 2022; Li et al., 2024b).

**Baselines.** We benchmark InertialAR against established models, including G-SchNet (Gebauer et al., 2019), ENF (Satorras et al., 2022a), EDM (Hoogeboom et al., 2022), GDM (Hoogeboom et al., 2022), EDM-Bridge (Wu et al., 2022), MiDi (Vignac et al., 2023), GeoLDM (Xu et al., 2023), UniGEM (Feng et al., 2025) and Geo2Seq (Li et al., 2024b).

**Results on QM9 and GEOM-Drugs.** Table 1 highlights the strong performance of InertialAR across both QM9 and GEOM-Drugs benchmarks. On QM9, InertialAR achieves the highest scores on Atom Stability (99.3%) and Molecule Stability (94.7%), surpassing all competing methods and indicating its ability to generate chemically consistent and structurally reliable molecules. On the larger and more complex GEOM-Drugs dataset, InertialAR continues to demonstrate superiority, attaining the best Atom Stability (87.2%) among all baselines. These results underscore the robustness of InertialAR in ensuring both local chemical validity and global structural stability, validating its effectiveness as a powerful autoregressive framework for 3D molecule generation.

**Results on B3LYP.** Due to the prohibitive computational cost of training all existing models on the large-scale B3LYP benchmark, we focus our comparison on two representa-

Table 2: Unconditional generation on 3D molecules on B3LYP-1M.

| | Valid (%) | Valid&Uni (%) | AtomSta (%) | MolSta (%) |
|---|---|---|---|---|
| EDM | 92.9 | 92.8 | 80.6 | 0.8 |
| Geo2Seq | 73.3 | 2.7 | 10.0 | 0.0 |
| InertialAR | **99.0** | **98.6** | **84.8** | **24.2** |

tive strong baselines: the diffusion-based EDM and the autoregressive Geo2Seq. The main results are shown in Table 2, InertialAR achieves substantial improvements over baselines on the large-scale B3LYP benchmark. Compared to the strong diffusion model EDM, it attains significantly higher validity and atom stability. Most notably, InertialAR shows a dramatic gain in Molecule Stability (24.2% vs. 0.8%), demonstrating its ability to produce chemically consistent molecules at scale. In contrast, the autoregressive baseline Geo2Seq performs poorly, highlighting the robustness and scalability of our approach on this chemically diverse dataset.

## 4.2 CLASS-CONDITIONAL 3D MOLECULE GENERATION AND MOLECULE EDITING

In chemistry and biology, class-conditional generation is particularly valuable, as "molecule classes" can correspond to key attributes such as chemical functionality, thereby enabling the targeted design or editing of molecules for drug discovery and materials science.

To enable class-conditional generation on QM9, we reconstruct the dataset by assigning each molecule a **Molecule Class ID** that encodes its functional group configuration (as shown in Figure 4). Specifically, we first convert each 3D structure to its SMILES string and then apply a rule-based SMARTS-matching system to detect predefined functional groups. The resulting presence/absence pattern is encoded as a binary string (*e.g.* "TTFFTFTT..."). Finally, through a predefined Functional Group Pattern-to-Class ID look-up, each molecule is assigned a corresponding Molecule Class ID.

The task is then to generate molecules conditioned on a specified functional group configuration. Concretely, we select the 5 most frequent Molecule Class IDs as conditioning targets. In addition to the metrics used for unconditional genera-

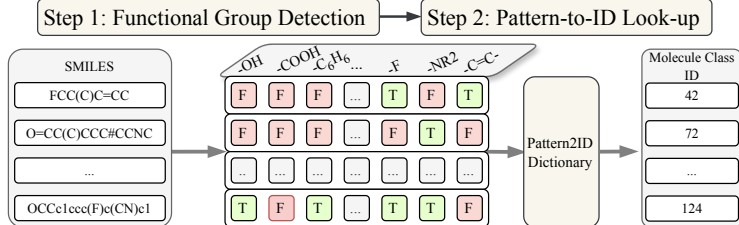

Figure 4: Overview of mapping 3D molecules to their Molecule Class IDs.

tion, we introduce a critical new metric for class-conditional generation, **Hit Rate**, which measures the proportion of generated molecules satisfying the target functional group requirements. A higher hit rate indicates stronger controllability of the generation process.

**Baselines.** We compare the conditional generation performance of InertialAR against the same representative autoregressive and diffusion-based baselines as in the unconditional setting, namely Geo2Seq and EDM, to ensure a consistent and fair comparison. **Results.** Table 3 shows that InertialAR achieves a remarkable average hit rate of 83.3%, significantly sur-

Table 3: Class-conditional generation on 3D molecules on QM9.

| Class ID ($c$) | Model | Rate (%) | Valid (%) | Valid&Uni (%) | AtomSta (%) | MolSta (%) |
|---|---|---|---|---|---|---|
| 7
(w/ Ether) | EDM | 37.5 | 84.8 | 84.2 | 96.3 | 52.9 |
| | Geo2Seq | 40.1 | 65.0 | 52.1 | 87.6 | 33.8 |
| | InertialAR | **90.9** | **99.0** | **92.8** | **99.7** | **97.5** |
| 28
(w/ Hydroxyl
& Ether) | EDM | 29.0 | 86.8 | 85.9 | 96.4 | 54.1 |
| | Geo2Seq | 44.2 | 64.7 | 55.6 | 86.5 | 33.4 |
| | InertialAR | **89.8** | **99.9** | **90.8** | **99.9** | **99.2** |
| 3
(w/ Hydroxyl) | EDM | 27.6 | 85.3 | 84.0 | 96.7 | 56.5 |
| | Geo2Seq | 49.4 | 70.3 | 53.9 | 89.7 | 42.2 |
| | InertialAR | **85.7** | **99.9** | **86.9** | **99.9** | **99.4** |
| 16
(w/ Heteroa-
romatic Ring) | EDM | 8.9 | 63.5 | 63.4 | 82.9 | 35.3 |
| | Geo2Seq | 33.8 | 65.6 | 57.8 | 86.4 | 34.8 |
| | InertialAR | **68.5** | **92.2** | **79.3** | **97.1** | **81.0** |
| 23
(w/ Secondary
Amine & Ether) | EDM | 25.3 | 76.8 | 76.7 | 96.1 | 53.3 |
| | Geo2Seq | 43.5 | 80.5 | 51.7 | 91.8 | 52.4 |
| | InertialAR | **81.8** | **99.7** | **82.7** | **99.9** | **99.2** |

passing EDM (25.7%) and Geo2Seq (42.2%), demonstrating its strong controllability in generating molecules that match the target functional group configurations. Beyond controllability, InertialAR also achieves excellent performance on chemical feasibility metrics, consistently outperforming both baselines across all evaluated molecule classes. These results highlight the effectiveness of InertialAR in producing both chemically valid and functionally precise molecules.

**Molecule Editing via CFG.** To further assess controllability, we examine the effect of varying the CFG guidance scale. Increasing the scale not only improves validity-related metrics but also enables molecule editing: molecules that originally lacked the required functional groups and exhibited unreasonable structures can be transformed to satisfy the target Molecule Class ID. As illustrated in Figure 3, by raising the guidance scale by 0.3 ($\Delta s = 0.3$), the generated molecules incorporate the desired functional groups while yielding more plausible 3D geometries, demonstrating that CFG enhances both structural validity and compliance with functional group constraints.

## 5 CONCLUSION

We propose InertialAR, a hierarchical autoregressive model that ensures SE(3) and permutation invariance through canonical tokenization while equipping Transformers with geometric awareness via GeoRoPE. This advances 3D molecule generation beyond restrictive physical priors and highlights its potential as a foundation model for scientific discovery.

**Future Directions.** InertialAR can be extended to more complex domains such as protein structure modeling and periodic material discovery, and can be integrated into broader multimodal frameworks, paving the way toward unified, AI-driven scientific discovery.

ETHICS STATEMENT

This research adheres fully to the ICLR Code of Ethics. It does not involve human subjects, personal information, or sensitive data. All datasets and code used or released comply with their respective licenses and terms of use. The contributions of this work are methodological and foundational, raising no concerns related to fairness, privacy, security, or potential misuse.

REPRODUCIBILITY STATEMENT

We are committed to ensuring the reproducibility of our results. Upon acceptance, we will release comprehensive resources on GitHub, including dataset access, experimental setup, model configurations, evaluation metrics, and checkpoints. Clear documentation and scripts will accompany these materials to enable accurate replication of all main results, thereby promoting transparency and scientific rigor.

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

## A    THE USE OF LARGE LANGUAGE MODELS

In this work, we employed large language models to refine English writing. All suggestions generated by the LLM were critically reviewed, vetted, and approved by the authors to ensure accuracy and integrity. The final manuscript faithfully represents the authors' own ideas, arguments, and research findings.

## B   Preliminaries and Related Work

### B.1   3D Molecule Generation

In the domain of AI-driven molecule discovery, 3D molecule generation has become a central problem. Its goal is to directly construct physically plausible 3D molecular conformations. Formally, a 3D molecule with $n$ atoms can be represented as a point cloud $G = (z, R)$. The vector $z = [z_1, \cdots, z_n] \in \mathbb{Z}^n$ encodes the atom types, where $z_i$ denotes the nuclear charge of the $i$-th atom. The coordinate matrix $R = [r_1, \cdots, r_n] \in \mathbb{R}^{3 \times n}$ specifies the 3D position of each atom, with $r_i \in \mathbb{R}^3$. A fundamental challenge lies in ensuring that molecular geometries respect the inherent SE(3) symmetry, i.e., molecular representations must remain invariant or equivariant under SE(3) transformations such as rotations and translations.

Current approaches can be categorized into four main paradigms. SE(3)-equivariant architectures explicitly enforce symmetry through specialized network designs: spherical frame basis models (Thomas et al., 2018b; Liao & Smidt, 2023) project features into irreducible representations of SO(3), while vector frame basis models (Satorras et al., 2022b; Schütt et al., 2021) construct local coordinate frames for equivariant operations. Invariant feature approaches circumvent architectural constraints by utilizing geometrically invariant inputs such as pairwise distances, bond angles, and dihedral angles (Schütt et al., 2017). Data augmentation strategies encourage models to implicitly learn symmetric representations by training on randomly rotated and translated molecular conformations, particularly valuable for large-scale models where explicit equivariance is complex to scale (Abramson et al., 2024). Input canonicalization methods (Li et al., 2024b; Fu et al., 2024) establish a canonical orientation or reference frame for input molecules through preprocessing, transforming each molecule into a standardized pose so that subsequent neural networks can operate on SE(3)-invariant inputs without intrinsic SE(3)-equivariant constraints.

A representative canonicalization strategy defines an inertial reference frame for each molecule using principal component analysis (PCA) (Guo et al., 2025; Lu et al., 2025a; Cheng et al., 2025a). After shifting the molecular coordinates so that the center of mass lies at the origin, the moment of inertia matrix is diagonalized to obtain the principal axes of rotation. Aligning the coordinates with these axes yields a canonical pose, unique up to axis reflections, effectively removing translational and rotational ambiguities. This inertial frame ensures SE(3)-symmetry molecular representations, enabling neural networks to process standardized and physically consistent 3D geometries without explicit equivariant design.

PCA-based inertial frames provide a simple and effective practical canonicalization strategy. Empirically, we find that PCA canonical poses are highly stable on real molecular datasets, making them an efficient SE(3) canonicalization choice for unconstrained architectures. Theoretically, however, PCA-based canonicalization is not strictly unique. Its limitations include potential axis flips from small geometric perturbations and ambiguity in axis orientation when principal moments are tied. These theoretical non-uniqueness issues have motivated a line of canonicalization-based symmetry handling methods that study how to systematically manage symmetry-equivalent frames. Frame Averaging (Puny et al., 2022) treats canonicalization as an equivariant projection by averaging outputs across all symmetry-equivalent PCA frames, while subsequent work shows that any finite, unweighted canonicalization procedure necessarily introduces discontinuities under symmetric configurations (Dym et al., 2024). More recent developments include Minimal Frame Averaging (Lin et al., 2024), which constructs theoretically minimal frames via stabilizer groups, and general canonicalization frameworks that reinterpret Frame Averaging and related strategies as orbit canonicalization (Ma et al., 2024). Our approach is complementary to this line: we adopt our canonical inertial frame as a simple and empirically robust canonicalization strategy, while these canonicalization-based methods provide principled tools that could further enhance robustness in future extensions.

### B.2   Autoregressive Models and Tokenization of 3D Molecule

Autoregressive models address sequence modeling by framing it as a "next-token prediction" problem. This approach, a direct application of the chain rule of probability, factorizes the joint distribu-

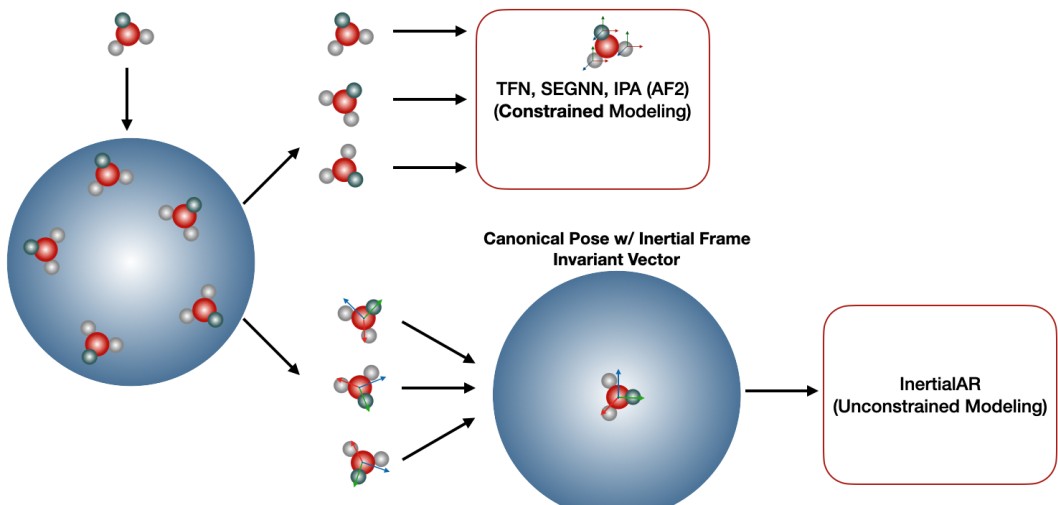

Figure 5: Comparison of existing SE(3)-equivariant graph neural networks and InertialAR.

tion of a sequence $x = (x_1, \ldots, x_n)$ into a product of conditional probabilities:

$$p(x) = p(x_1, \ldots, x_n) = \prod_{i=1}^{n} p(x_i | x_1, \ldots, x_{i-1}).$$

The model's core task is thus to learn the conditional distribution $p(x_i|x_{<i})$ for each step, which is typically parameterized by a powerful neural network such as Transformer. The primary challenge in applying autoregressive models to 3D molecular generation lies in the effective **structure tokenization** of a 3D molecular structure into a 1D sequence of tokens suitable for Transformer architectures. The choice of tokenization strategy is crucial, as it defines not only the sequence representation but also the very nature of the conditional modeling itself. Existing approaches can be broadly classified into three main categories:

**Voxel-based tokenization**, which discretizes the 3D space occupied by a molecule into a 3D grid, draws a direct parallel to image generation (Faltings et al., 2025; Lu et al., 2025b). Each voxel in the grid serves as a token that encodes local atomic information, much like a pixel in an image. **Text sequence-based tokenization**, which is similar to language modeling, serializes 3D molecules into a 1D, text-like sequence (Li et al., 2024b; Yan et al., 2024; Flam-Shepherd & Aspuru-Guzik, 2023). The process involves discretizing continuous 3D coordinates and concatenating them with discrete atom types. This treats a molecule like a sentence, where every atom type and 3D coordinates are encoded as words. **Atom-based tokenization** directly treats an atom as one single token that encapsulates both its discrete atom type and continuous 3D coordinates. This establishes an intuitive correspondence between the physical atoms and their tokenized representation, thereby preserving atom-level granularity.

### B.3 CLASS-CONDITIONAL GENERATION AND CLASSIFIER-FREE GUIDANCE

Class-conditional generation is a paradigm that generates samples conditioned on a specific class label $c$. In image generation, this involves generating an image guided by a prefix class embedding (Esser et al., 2021; Peebles & Xie, 2023). In chemistry and biology, class-conditional generation is highly useful, as molecular "classes" can correspond to key attributes such as chemical functionality or physicochemical characteristics, enabling the targeted design or editing of molecules for drug discovery and materials science.

Classifier-free guidance (CFG) improves both sample quality and fidelity to conditions by randomly dropping conditioning signals during training (Ho & Salimans, 2022). This simple yet effective strategy enables a single model to jointly learn both the conditional distribution $p(x|c)$ and the unconditional distribution $p(x)$. At inference, the difference between these two learned distributions is then leveraged to amplify the conditional signal without relying on an auxiliary classifier. Although

originally proposed for diffusion, CFG has also proven effective in autoregressive image generation, showing great potential for molecule generation.

## B.4 DIFFUSION LOSS FOR AUTOREGRESSIVE MODELS

While autoregressive models are naturally suited for generating discrete tokens via cross-entropy loss, 3D molecule generation introduces an additional challenge: predicting continuous 3D coordinates. Diffusion Loss (Li et al., 2024a) provides an effective framework to extend autoregressive models to continuous-valued token generation. Formally, to predict the continuous-valued token $x_i$, the autoregressive model first outputs a vector $h_{i-1}$ conditioned on previous tokens $x_{<i}$. The objective is to model the conditional probability distribution $p(x_i|h_{i-1})$. Diffusion loss achieves this through a denoising score-matching objective:

$$L(x_i, h_{i-1}) = \mathbb{E}_{\epsilon,t} \left[ |\epsilon - \epsilon_\theta(x_i^t|t, h_{i-1})|^2 \right], \tag{20}$$

where $x_i^t = \sqrt{\bar{\alpha}_t}x + \sqrt{1-\bar{\alpha}}t\epsilon$ is a noised version of $x_i$, and $\epsilon_\theta$ a denoising network that predicts the noise $\epsilon$ conditioned on $\mathbf{z}$ and timestep $t$. Gradients from this loss propagate through $h_{i-1}$, enabling end-to-end training of the autoregressive backbone.

This approach preserves the strong sequence modeling capacity of autoregressive models while extending them to predict continuous distributions. By directly modeling 3D coordinates, it removes the need for discretization or coarse tokenization of molecular geometries and provides a principled mechanism for generating chemically precise molecular structures.

# C    CLASS-CONDITIONAL GENERATION

## C.1    DATASET RECONSTRUCTION

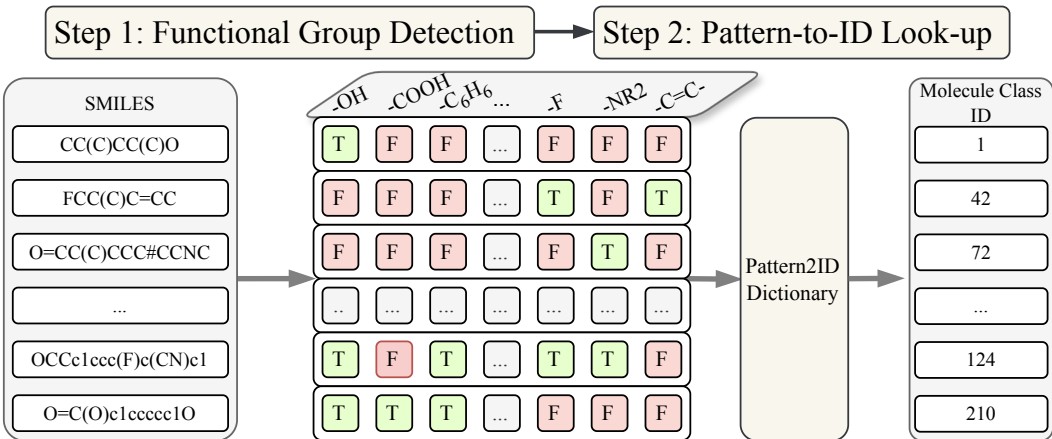

Figure 6: Overview of how 3D molecules are mapped to their Molecule Class IDs.

In chemistry and biology, class-conditional generation is highly useful, as "molecule classes" can correspond to key attributes such as chemical functionality or physicochemical characteristics, enabling the targeted design or editing of molecules for drug discovery and materials science. However, commonly used datasets, such as QM9 and Geom-Drug, do not provide explicit functional group annotations. To enable controllable molecule generation with specified functional group configurations, we reconstruct the datasets by assigning each molecule a unique class label (Molecule Class ID) that encodes its functional group composition. Concretely, we design a comprehensive labeling pipeline based on functional groups (shown in Figure 6): for each molecule, we first convert its 3D structure to a SMILES representation. We then employ a rule-based system with a library of SMARTS queries to identify the presence or absence of a predefined set of functional groups. The resulting pattern is encoded as a binary string (e.g., "TTFFTFTT..."), where each position indicates the presence (T) or absence (F) of a functional group. Finally, through a predefined Functional Group Pattern-to-Class ID mapping, each molecule is assigned a corresponding Molecule Class ID.

## C.2    CONTROLLABLE GENERATION WITH CLASSIFIER-FREE GUIDANCE

Originally developed in the diffusion model community, classifier-free guidance (CFG) is widely recognized for improving both sample quality and conditional alignment. The key idea is to train a single model that jointly learns the conditional distribution $p(x|c)$ and the unconditional distribution $p(x)$ by randomly dropping conditioning labels during training.

We incorporate classifier-free guidance (CFG, details in Section 2) into InertialAR to enable controllable generation. During inference, CFG modulates conditional generation process by leveraging the difference between conditional and unconditional predictions:

$$p_g = p_u + s(p_c - p_u), \tag{21}$$

where $p_c$ and $p_u$ denote the conditional and unconditional predictions, respectively, and $s$ is the guidance scale. In InertialAR, CFG is applied to the estimated noise $\epsilon_\theta$ in diffusion for coordinates generation, as well as to the logits over a discrete vocabulary for atom type prediction. By tuning $s$, we can achieve both stronger adherence to target molecular classes and better structural validity.

## D  NYSTRÖM ESTIMATION

The Nyström method (Williams & Seeger, 2000) is a low-rank approximation to obtain the pairwise distance. More concretely, suppose we have a Gram matrix over $n$ points, *i.e.*, $K \in \mathbb{R}^{n \times n}$. Each element $K_{ij}$ is the radial basis function (RBF) over the distance between $i$-th and $j$-th points, $K_{ij} = RBF(x_i, x_j) = \exp(-\frac{\|x_i - x_j\|^2}{2\sigma^2})$. Then we sample $m$ anchor points, $(\boldsymbol{c}_1, \boldsymbol{c}_2, ..., \boldsymbol{c}_m)$, where $\boldsymbol{c}$ is 3D coordinates in an Euclidean space and $n \gg m$.

First, we can decompose the matrix $K$ with eigendecomposition,

$$K = U \Lambda U^T, \tag{22}$$

where $U \in \mathbb{R}^{n \times n}$ is an orthogonal matrix whose columns are the orthonormal eigenvectors of $K$, and $\Lambda \in \mathbb{R}^{n \times n}$ is a diagonal matrix whose entries are the corresponding eigenvalues of $K$.

Then, Nyström approximation is a low rank approximation, assuming that matrix $K$ can be approximated using $\tilde{K}$:

$$
\begin{aligned}
K &\approx \tilde{K} \\
&= \tilde{U} \tilde{\Lambda} \tilde{U}^T \\
&= \begin{bmatrix} A & B \\ B^T & C \end{bmatrix},
\end{aligned} \tag{23}
$$

where $\tilde{U}$ is the first $m$ columns of $U$ and $\tilde{\Lambda}$ is the block diagonal matrix of first $m$ eigenvalues of $\Lambda$. At this point, we assume that the $m$ points picked can estimate the $m$-rank matrix $A$ with positive eigenvalues. Then let us have $\tilde{U} = \begin{bmatrix} \tilde{U}_1 \\ \tilde{U}_2 \end{bmatrix}$, where $\tilde{U}_1 \in \mathbb{R}^{m \times m}$ and $\tilde{U}_2 \in \mathbb{R}^{(n-m) \times m}$. This means $A = \tilde{U}_1 \tilde{\Lambda} \tilde{U}_1^T$ and $B = \tilde{U}_2 \tilde{\Lambda} \tilde{U}_2$. Thus, we can rewrite Equation (23) as:

$$
\begin{aligned}
\tilde{K} &= \begin{bmatrix} \tilde{U}_1 \\ \tilde{U}_2 \end{bmatrix} \tilde{\Lambda} \begin{bmatrix} \tilde{U}_1 \\ \tilde{U}_2 \end{bmatrix}^T \\
&= \begin{bmatrix} \tilde{U}_1 \tilde{\Lambda} \tilde{U}_1^T & \tilde{U}_1 \tilde{\Lambda} \tilde{U}_2^T \\ \tilde{U}_2 \tilde{\Lambda} \tilde{U}_1^T & \tilde{U}_2 \tilde{\Lambda} \tilde{U}_2^T \end{bmatrix}.
\end{aligned} \tag{24}
$$

Combining this with Equation (23), we have $\tilde{U}_2 = B^T \tilde{U}_1 \tilde{\Lambda}^{-1}$ and $\tilde{U}_2^T = \tilde{\Lambda}^{-1} \tilde{U}_1^T B$. Thus, we can have

$$C = \tilde{U}_2 \tilde{\Lambda} \tilde{U}_2^T = B^T \tilde{U}_1 \tilde{\Lambda}^{-1} \tilde{U}_1^T B = B^T A^{-1} B. \tag{25}$$

To inject this back to Equation (23), we have

$$
\begin{aligned}
\tilde{K} &= \begin{bmatrix} A & B \\ B^T & B^T A^{-1} B \end{bmatrix} \\
&= \begin{bmatrix} A \\ B^T \end{bmatrix} A^{-1} \begin{bmatrix} A & B \end{bmatrix}.
\end{aligned} \tag{26}
$$

This wraps up the key idea of Nyström method. Then, to obtain the RBF of a new point pair $K(i, j)$, we first construct the feature between point $i, j$ and the $m$ anchor points as $k_i = [\text{RBF}(i, 0), \text{RBF}(i, 1), ..., \text{RBF}(i, m)]^T \in \mathbb{R}^{m \times 1}$. The approximated $\text{RBF}(i, j)$ can be obtained as:

$$
\begin{aligned}
\tilde{k}(i, j) &= k_i^T A^{-1} k_j \\
&= \left(A^{-1/2} k_i\right)^T \left(A^{-1/2} k_j\right) \\
&= \left(L^{-1} k_i\right)^T \left(L^{-1} k_j\right),
\end{aligned} \tag{27}
$$

where $A = LL^T$ is the Cholesky decomposition.

For each atom $i$, we define its Nyström Approximation Encoding as

$$z_i^{\text{Nyström}} = L^{-1} k_i. \tag{28}$$

This design allows the approximated RBF, which encodes the pairwise distance information between atoms, to be recovered directly by the inner product within the attention mechanism:

$$\tilde{k}(i,j) = (z_i^{\text{Nyström}})^T(z_j^{\text{Nyström}}). \tag{29}$$

**Discussion.** There is another research line using random features (*e.g.*, random Fourier features) for the pairwise distance approximation (Rahimi & Recht, 2007). There are certain works that have proved that Nyström method is more accurate (Yang et al., 2012). One intuitive way to understand this is that Nyström method utilizes the data-dependent basis, while the random features use data-independent basis functions.

# E    DETERMINE INERTIAL FRAME DIRECTIONS BY INTRODUCING FOURTH NODE

**Theorem 2.** *For an inertial frame $F$, we build up the corresponding right-handed axes as coordinate systems $Q$. Then we need to incorporate a fourth point that is coplanar with the three basis vectors to uniquely determine the directions of the coordinate system with one rotation transformation matrix.*

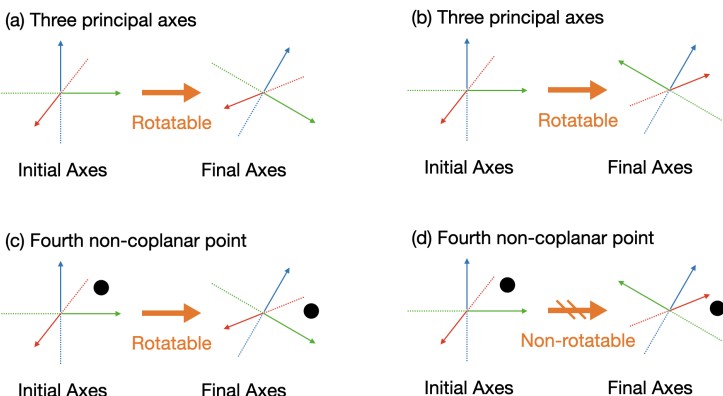

Figure 7: (a, b) show two potential rotational alignments between two coordinate systems (axes). (c, d) show that only one unique rotation is possible for four non-coplanar points.

*Proof.* For three vectors, we can easily find a counter-example , as illustrated in Figure 7 (a, b). Figure 7 (a, b) describes two cases where we have the same initial frame, and we can rotate it to two different final frames with two rotation matrices, yet the righthandness still matches. We can easily see that there are four options of rotation matrices in this case, and we cannot uniquely determine the final inertial frame in this case.

More rigorously, let us first assume that there exists a rotation transformation $R$ that can transform the initial coordinate system $Q_i$ to the final coordinate system $Q_f$, as:

$$
\begin{bmatrix} Q_{f,0} \\ Q_{f,1} \\ Q_{f,2} \end{bmatrix}^T = R \cdot \begin{bmatrix} Q_{i,0} \\ Q_{i,1} \\ Q_{i,2} \end{bmatrix}^T
\tag{30}
$$

First, we should change either zero or two directions for direction alignment. Then, without loss of generality, we can assume the two directions to be the last two axes. Thus, we can obtain a rotation matrix $R'$ such that $R'$ is rotating $R$ along vector $Q_{f,0}$ with 180 degrees. We can represent $R'$ using Rodrigue's rotation formula, as $R' = (2Q_{f,0}Q_{f,0}^T - I)R$. Thus, we can have:

$$
R' \cdot \begin{bmatrix} Q_{i,0} \\ Q_{i,1} \\ Q_{i,2} \end{bmatrix}^T = (2Q_{f,0}Q_{f,0}^T - I) \begin{bmatrix} Q_{f,0} \\ Q_{f,1} \\ Q_{f,2} \end{bmatrix}^T = \begin{bmatrix} Q_{f,0} \\ -Q_{f,1} \\ -Q_{f,2} \end{bmatrix}^T
\tag{31}
$$

This is essentially saying starting from one initial frame, we can have multiple matched final frames. Thus, using only three vectors cannot uniquely determine the direction matching. We provide two examples in Figure 7 (a, b).

For the four vectors, we introduce an extra atom into the inertial frame system, and such an extra atom point is nonplanar to the three base axes. Then the problem becomes: starting from an initial frame and an extra point, can we find multiple rotation matrices such that the final frames have reflected directions? To be more rigorous, let us have the following formulation.

First, let us assume we have this rotation matrix:

$$
\begin{bmatrix} Q_{f,0} \\ Q_{f,1} \\ Q_{f,2} \\ \boldsymbol{v} \end{bmatrix}^T = R \cdot \begin{bmatrix} Q_{i,0} \\ Q_{i,1} \\ Q_{i,2} \\ \boldsymbol{v} \end{bmatrix}^T
\tag{32}
$$

As discussed above, we need to guarantee the right-handedness property, thus, without loss generality, here we also assume the last two axes are reflected. The question turns to: does it exit another rotation matrix $R'$, such that:

$$
\begin{bmatrix} Q_{f,0} \\ -Q_{f,1} \\ -Q_{f,2} \\ \boldsymbol{v} \end{bmatrix}^T = R' \cdot \begin{bmatrix} Q_{i,0} \\ Q_{i,1} \\ Q_{i,2} \\ \boldsymbol{v} \end{bmatrix}^T \tag{33}
$$

We now use contradiction. Since we still have the two axes rotated 180 degrees around the first axes, $Q_{f,0}$, so $R' = (2Q_{f,0}Q_{f,0}^T - I)R$. Then given the two conditions $\boldsymbol{v}^T = R\boldsymbol{v}^T$ and $\boldsymbol{v}^T = R'\boldsymbol{v}^T$, we have $(2Q_{f,0}Q_{f,0}^T - I)\boldsymbol{v}^T = \boldsymbol{v}^T$.

If we let $Q_{f,0} = [k_1, k_2, k_3]$ and $\boldsymbol{v} = [v_1, v_2, v_3]$, then we have

$$
(2Q_{f,0}Q_{f,0}^T - I)\boldsymbol{v}^T = \boldsymbol{v}^T
$$

$$
\begin{bmatrix} k_1 k_1 & k_1 k_2 & k_1 k_3 \\ k_1 k_2 & k_2 k_2 & k_2 k_3 \\ k_2 k_3 & k_2 k_3 & k_3 k_3 \end{bmatrix} \begin{bmatrix} v_1 \\ v_2 \\ v_3 \end{bmatrix} = \begin{bmatrix} v_1 \\ v_2 \\ v_3 \end{bmatrix}
$$

$$
\begin{bmatrix} k_1(k_1 v_1 + k_2 v_2 + k_3 v_3) \\ k_2(k_1 v_1 + k_2 v_2 + k_3 v_3) \\ k_3(k_1 v_1 + k_2 v_2 + k_3 v_3) \end{bmatrix} = \begin{bmatrix} v_1 \\ v_2 \\ v_3 \end{bmatrix} \tag{34}
$$

$$
(k_1 v_1 + k_2 v_2 + k_3 v_3) \begin{bmatrix} k_1 \\ k_2 \\ k_3 \end{bmatrix} = \begin{bmatrix} v_1 \\ v_2 \\ v_3 \end{bmatrix} .
$$

After calculation, we can obtain that $Q_{f,0} = c\boldsymbol{v}$, where $c$ is a coefficient. However, as we claimed in the condition, $\boldsymbol{v}$ does not lie in the same line as $Q_{f,0}$, thus, there does not exist such another rotation matrix $R' \neq R$ satisfying Equation (33). We also provide two examples in Figure 7 (c, d).

By contradiction, we can tell that there is only one unique rotation mapping from the initial inertial frame to the final inertial frame. □

To sum up, three points cannot form a rigid structure in Euclidean space, thus there can exist multiple reflection transformations, leading to opposite inertial frames. Four points can form a rigid structure, thus there exists only one reflection transformation.

# F  POSITIONAL ENCODING

Position embedding is one of the most important building blocks in Transformer. There have been multiple methods, and we would like to briefly discuss them here.

## F.1  ABSOLUTE POSITIONAL ENCODING

In absolute positional encoding, each position in the input sequence is assigned a unique, fixed embedding. The most classical positional encoding is the sinusoidal function Vaswani (2017):

$$\begin{cases} p_{i,2t} = \sin(k/10000^{2t/d}) \\ p_{i,2t+1} = \cos(k/10000^{2t/d}) \end{cases} \tag{35}$$

Such a positional encoding will be added (or multiplied) to the token embedding, and the classical attention module is as:

$$\begin{aligned} q_i &= (x_i + p_i)W_Q \\ k_j &= (x_j + p_j)W_K \\ v_j &= (x_j + p_j)Q_V \\ a_{i,j} &= \text{softmax}(q_k k_j^T) \\ o_i &= \sum_j a_{i,j} v_j. \end{aligned} \tag{36}$$

Pros:

- Simple and easy to implement.
- Provides a clear, ordered embedding that the model can use to distinguish between different token positions.

Cons:

- Limited in capturing the relative distance between tokens, especially in very long sequences.
- Fixed nature can limit the model's ability to generalize to longer sequences beyond what it was trained on.

## F.2  RELATIVE POSITIONAL ENCODING

In relative positional encoding, the model encodes the distance (or relative position) between tokens rather than absolute positions. The relative distance will be further used in calculating the attention score.

The first relative positional encoding was proposed in Shaw et al. (2018), as:

$$\begin{aligned} R_{i,j}^K &= p_K[\text{clip}(i-j, p_min, p_max)] \\ R_{i,j}^V &= p_V[\text{clip}(i-j, p_min, p_max)], \end{aligned} \tag{37}$$

where $p_K, p_V$ are certain learnable functions or non-learnable functions (like sinusoidal function in Equation (35)). This will be then used to define the attention score, which will be then replaced to Equation (36):

$$a_{i,j} = \text{softmax}(x_i W_Q(x_j W_K + R_{i,j}^K)^T) \tag{38}$$

Based on this, there are more variants on defining the relative distance, such as XLNet Dai (2019), DeBERTa He et al. (2020), and T5 Raffel et al. (2020).

Pros:

- More flexible and generalizable, especially to unseen sequence lengths.
- Better at capturing the local context by focusing on distances between tokens.

Cons:

- Can be more complex to implement and computationally intensive.
- The model might need to adapt if positional relationships are nuanced.

### F.3 ROTARY POSITIONAL ENCODING: HYBRID OF ABSOLUTE & RELATIVE POSITIONAL ENCODING

Rotary Positional Encoding (RoPE) (Su et al., 2024) is a hybrid of both the absolute and relative positional encoding.

The core idea is the utilization of complex numbers and their inner-product property. More concretely, we have the query and key vectors as $q_m e^{im\theta}$ and $k_n e^{in\theta}$, and their inner product is:

$$
\begin{aligned}
\langle q_m e^{im\theta}, k_n e^{in\theta} \rangle &= \mathrm{Re}\big[ (q_m e^{im\theta}) \overline{(k_n e^{in\theta})} \big] \\
&= \mathrm{Re}\big[ q_m \overline{k_n} \, e^{i(m-n)\theta} \big],
\end{aligned}
\tag{39}
$$

where $\mathrm{Re}[\cdot]$ denotes the real part and $\overline{(\cdot)}$ denotes complex conjugation. Here $q_m$ and $k_n$ denote the complex-valued 2D sub-block representations (paired components) of the original real vectors.

**RoPE As Absolute Positional Encoding:** RoPE uses absolute positions to determine each token's rotation, hence capturing absolute positional information directly in each embedding.

**RoPE As Relative Positional Encoding:** The relative distances are captured in the attention layer, thanks to the inner product of rotated embeddings, which varies based on the distance between positions. This allows RoPE to behave similarly to relative positional encodings in the self-attention scores.

**Matrix Format:** Considering Euler's formula $e^{i\theta} = \cos\theta + i\sin\theta$, we can further write this in the following formation for matrix $q$ with even dimension $d$:

$$
\underbrace{\begin{bmatrix}
\cos m\theta_0 & -\sin m\theta_0 & 0 & 0 & \cdots & 0 & 0 \\
\sin m\theta_0 & \cos m\theta_0 & 0 & 0 & \cdots & 0 & 0 \\
0 & 0 & \cos m\theta_1 & -\sin m\theta_1 & \cdots & 0 & 0 \\
0 & 0 & \sin m\theta_1 & \cos m\theta_1 & \cdots & 0 & 0 \\
\vdots & \vdots & \vdots & \vdots & \ddots & \vdots & \vdots \\
0 & 0 & 0 & 0 & \cdots & \cos m\theta_{d/2-1} & -\sin m\theta_{d/2-1} \\
0 & 0 & 0 & 0 & \cdots & \sin m\theta_{d/2-1} & \cos m\theta_{d/2-1}
\end{bmatrix}}_{R_m}
\begin{bmatrix}
q_0 \\ q_1 \\ q_2 \\ q_3 \\ \vdots \\ q_{d-2} \\ q_{d-1}
\end{bmatrix}
\tag{40}
$$

We can easily observe that:

$$
(R_m q)^T (R_n k) = q^T R_m^T R_n k = q^T R_{n-m} k.
\tag{41}
$$

**Preliminary Matrix Exponential Format:** First, let us recall that for a complex number, $z = a + bi$, we can write it in the following formats:

$$
\begin{aligned}
z &= a + bi \\
&= r\cos\theta + ir\sin\theta \\
&= re^{i\theta} \\
&= r \begin{bmatrix} \cos\theta & -\sin\theta \\ \sin\theta & \cos\theta \end{bmatrix}.
\end{aligned}
\tag{42}
$$

Here we want to prove that this complex number can be further written in a matrix exponential way.

$$
\begin{aligned}
r \exp\left(\theta \begin{bmatrix} 0 & -1 \\ 1 & 0 \end{bmatrix}\right) &\equiv r\exp(\theta J) \\
&= r \sum_{n=0}^{\infty} \frac{(\theta J)^n}{n!}.
\end{aligned}
\tag{43}
$$

*Proof.* Because matrix $J$ has an interesting property: $J^2 = -I$, $J^3 = -J$, $J^4 = I$, thus we can rewrite as:

$$
\begin{aligned}
r \exp(\theta J) &= r \sum_{n=0}^{\infty} \frac{(\theta J)^n}{n!} \\
&= r\left(I + \theta J - \frac{\theta^2}{2!}I - \frac{\theta^3}{3!}J + \frac{\theta^4}{4!}I + \frac{\theta^5}{5!}J - \frac{\theta^6}{6!}I - \frac{\theta^7}{7!}J + \cdots\right) \\
&= rI \cdot \sum_{n=0}^{\infty} \frac{(-1)^n \theta^{2n}}{(2n)!} + rJ \cdot \sum_{n=0}^{\infty} \frac{(-1)^n \theta^{2n+1}}{(2n+1)!} \\
&= rI \cos\theta + rJ \sin\theta \\
&= r \begin{bmatrix} \cos\theta & -\sin\theta \\ \sin\theta & \cos\theta \end{bmatrix}.
\end{aligned}
\tag{44}
$$

$\square$

To sum up, for a complex number, we have:

$$
\begin{aligned}
z &= a + bi \\
&= r\cos\theta + ir\sin\theta \\
&= re^{i\theta} \\
&= r \begin{bmatrix} \cos\theta & -\sin\theta \\ \sin\theta & \cos\theta \end{bmatrix} \\
&= r \exp\left(\theta \begin{bmatrix} 0 & -1 \\ 1 & 0 \end{bmatrix}\right).
\end{aligned}
\tag{45}
$$

**Matrix Exponential Format for Rotary Positional Encoding:** Then we would like to explore how to incorporate this into rotary embedding. For the $2 \times 2$ rotation matrix $\begin{bmatrix} \cos m\theta & -\sin m\theta \\ \sin m\theta & \cos m\theta \end{bmatrix}$, we can write its matrix exponential form as $\exp(m\theta J)$, where $J = \begin{bmatrix} 0 & -1 \\ 1 & 0 \end{bmatrix}$.

Thus, our rotation matrix can be written as $R_n = \exp(n\theta J)$, and we can observe that

$$
\begin{aligned}
R_m^T R_n &= \exp(m\theta J)^T \exp(n\theta J) \\
&= \exp(-m\theta J) \exp(n\theta J) \\
&= \exp((n-m)\theta J) \\
&= R_{n-m}.
\end{aligned}
\tag{46}
$$

# G COMPLEX, QUATERNION, AND ROTATION

## G.1 COMPLEX

Any complex number $z \in \mathbb{C}$ can be written as $z = a + bi$, where $a, b \in \mathbb{R}$ and $i^2 = -1$. We call $a$ as the real part and $b$ as the imaginary part.

We can write $z$ as a vector, meaning the linear combination over the basis $\{1, i\}$:

$$z = \begin{bmatrix} a \\ b \end{bmatrix}. \tag{47}$$

**Addition** If have two complex numbers $z_1 = a + bi$ and $z_2 = c + di$, then the addition of two numbers is:

$$z_1 + z_2 = (a + c) + (b + d)i. \tag{48}$$

**Multiplication** If have two complex numbers $z_1 = a + bi$ and $z_2 = c + di$, then the multiplication of two numbers is:

$$\begin{aligned} z_1 z_2 &= (a + bi)(c + di) \\ &= (ac - bd) + (ad + bc)i. \end{aligned} \tag{49}$$

Or we can write this in a matrix-vector multiplication:

$$z_1 z_2 = \begin{bmatrix} a & -b \\ b & a \end{bmatrix} \begin{bmatrix} c \\ d \end{bmatrix}, \tag{50}$$

where $\begin{bmatrix} a & -b \\ b & a \end{bmatrix}$ is the matrix for $z_1$ and $\begin{bmatrix} c \\ d \end{bmatrix}$ is the vector for $z_2$.

**Conjugate** The conjugate of $z = a + bi$ is:

$$\bar{z} = a - bi. \tag{51}$$

## G.2 COMPLEX AND ROTATION

Multiplying a complex number $z = a + bi$ is equivalent to multiplying the matrix $\begin{bmatrix} a & -b \\ b & a \end{bmatrix}$, then the question is what does this matrix mean?

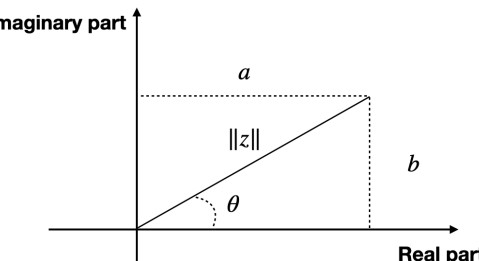

Figure 8: Illustration of the geometric representation of a complex number.

If we transform the matrix as follows:

$$\begin{aligned} \begin{bmatrix} a & -b \\ b & a \end{bmatrix} &= \sqrt{a^2 + b^2} \begin{bmatrix} \frac{a}{\sqrt{a^2+b^2}} & \frac{-b}{\sqrt{a^2+b^2}} \\ \frac{b}{\sqrt{a^2+b^2}} & \frac{a}{\sqrt{a^2+b^2}} \end{bmatrix} \\ &= \|z\| \begin{bmatrix} \cos(\theta) & -\sin(\theta) \\ \sin(\theta) & \cos(\theta) \end{bmatrix}. \end{aligned} \tag{52}$$

Then we can see that this matrix is indeed the rotation transformation on the 2D plane, as shown in Figure 8.

**Inner product or Hermitian inner product or conjugate symmetric inner product**   If have two complex numbers $z_1 = a + bi$ and $z_2 = c + di$, then the inner product of two numbers is:

$$
\begin{aligned}
\langle z_1, z_2 \rangle &= z_1 \bar{z}_2 \\
&= (a + bi) * (c - di) = ac + bd + (bc - ad)i \\
&= |z_1| e^{i\theta_1} \cdot |z_2| e^{-i\theta_2} = |z_1||z_2| e^{i(\theta_1 - \theta_2)} \\
&= |z_1|(\cos(\theta_1) + i \sin(\theta_1)) \cdot |z_2|(\cos(\theta_2) - i \sin(\theta_2)) \\
&= |z_1||z_2|(\cos(\theta_1 - \theta_2) + i \sin(\theta_1 - \theta_2)) \\
&= |z_1| \begin{bmatrix} \cos(\theta_1) \\ \sin(\theta_1) \end{bmatrix} \cdot |z_2| \begin{bmatrix} \cos(\theta_2) \\ -\sin(\theta_2) \end{bmatrix} \\
&= |z_1||z_2| \begin{bmatrix} \cos(\theta_1 - \theta_2) \\ \sin(\theta_1 - \theta_2) \end{bmatrix} \\
&= |z_1| \begin{bmatrix} \cos(\theta_1) & -\sin(\theta_1) \\ \sin(\theta_1) & \cos(\theta_1) \end{bmatrix} \cdot |z_2| \begin{bmatrix} \cos(\theta_2) & \sin(\theta_2) \\ -\sin(\theta_2) & \cos(\theta_2) \end{bmatrix} \\
&= |z_1||z_2| \begin{bmatrix} \cos(\theta_1 - \theta_2) & -\sin(\theta_1 - \theta_2) \\ \sin(\theta_1 - \theta_2) & \cos(\theta_1 - \theta_2) \end{bmatrix}.
\end{aligned}
\tag{53}
$$

### G.3   QUATERNION

A quaternion is defined as:

$$
q = a + bi + cj + dk,
\tag{54}
$$

where $a, b, c, d \in \mathbb{R}$ and $i^2 = j^2 = k^2 = ijk = -1$.

Similarly, we can also write quaternion as a vector, *i.e.*, the linear combination of basis $\{1, i, j, k\}$:

$$
q = \begin{bmatrix} a \\ b \\ c \\ d \end{bmatrix}.
\tag{55}
$$

We can rewrite this as:

$$
\begin{aligned}
q &= [w, \boldsymbol{u}] \\
&= |q|(\cos(\theta) + \sin(\theta)\boldsymbol{u}) \\
&= |q| e^{\theta \boldsymbol{u}} \\
&= |q|(\cos \theta + (x \sin \theta)i + (y \sin \theta)j + (z \sin \theta)k),
\end{aligned}
\tag{56}
$$

where $\boldsymbol{u} = \frac{xi + yj + zk}{\sqrt{x^2 + y^2 + z^2}}$.

**Addition**   If we have two quaternions $q_1 = a + bi + cj + dk$ and $q_2 = e + fi + gj + hk$, then the addition of two quaternions is:

$$
q_1 + q_2 = (a + e) + (b + f)i + (c + g)j + (d + h)k.
\tag{57}
$$

**Multiplication**   If we have two quaternions $q_1 = a + bi + cj + dk$ and $q_2 = e + fi + gj + hk$, then the multiplication of two quaternions is:

$$
\begin{aligned}
q_1 q_2 &= (a + bi + cj + dk)(e + fi + gj + hk) \\
&= ae + afi + agj + ahk + \\
&\quad bei + bfi^2 + bgij + bhij + \\
&\quad cej + cfji + cgj^2 + chjk + \\
&\quad dek + dfji + dgkj + dhk^2 \\
&= (ae - bf - cg - dh) + \\
&\quad (be + af - dg + ch)i + \\
&\quad (ce + df + ag - bh)j + \\
&\quad (de - cf + bg + ah)k \\
&= \begin{bmatrix} a & -b & -c & -d \\ b & a & -d & c \\ c & d & a & -b \\ d & -c & b & a \end{bmatrix} \begin{bmatrix} e \\ f \\ g \\ h \end{bmatrix}.
\end{aligned}
\tag{58}
$$

**Inner product**

$$
\langle q_1, q_2 \rangle = Re(q_1 * \bar{q}_2)
\tag{59}
$$

$$
\begin{aligned}
q_1 * \bar{q}_2 &= |q_1||q_2|(\cos(\theta_1) + \sin(\theta_1)\boldsymbol{u}_1)(\cos(\theta_2) + \sin(\theta_2) - \boldsymbol{u}_2) \\
&= [st + \boldsymbol{u}_1\boldsymbol{u}_2, s\boldsymbol{u}_2 + t\boldsymbol{u}_1 + \boldsymbol{u}_1 \times \boldsymbol{u}_2]
\end{aligned}
\tag{60}
$$

### G.4   QUATERNION AND ROTATION

First, we can have the quaternion to rotation matrix as:

$$
R(q) = \begin{bmatrix} 1 - 2y^2 - 2z^2 & 2xy - 2zy & 2xz + 2yw \\ 2xy + 2zw & 1 - 2x^2 - 2z^2 & 2yz - 2xw \\ 2xz - 2yw & 2yz + 2xw & 1 - 2x^2 - 2y^2 \end{bmatrix}
\tag{61}
$$

### G.5   TOKENIZATION

We assume that we would like to use the following equations to add absolute positions to $q$ and $k$:

$$
\tilde{q}_m = f(q, x_m, y_m, z_m), \quad \tilde{k}_n = f(k, x_n, y_n, z_n).
\tag{62}
$$

In other words, we hope that we can add the absolute position into $\tilde{q}$ and $\tilde{k}$.

Because the core module of attention is the inner product, so we prefer the following:

$$
\langle q_{x_m, y_m, c_m}, k_{x_n, y_n, c_n} \rangle = g(q, k, d_{m,n}),
\tag{63}
$$

where $d_{m,n} = \sqrt{(x_m - x_n)^2 + (y_m - y_n)^2 + (z_m - z_n)^2}$.

# H ABLATION STUDIES

In this section, we provide additional ablation studies and robustness analyses that were conducted during the rebuttal phase. Unless otherwise stated, all experiments are performed on the QM9 unconditional generation setting, and we report the same four metrics as in the main paper: Valid, Valid&Unique, AtomSta, and MolSta.

## H.1 ROBUSTNESS OF THE CANONICAL INERTIAL FRAME

We perform two complementary analyses to assess the robustness of the canonical inertial frame: (i) stability under small geometric perturbations, and (ii) frequency of principal-moment degeneracy in realistic datasets.

**Stability under small perturbations.** We add i.i.d. Gaussian noise to atomic coordinates in QM9 and Drugs to quantify how often the "farthest atom" (used for axis-sign resolution) changes. Since quantum-derived coordinates are typically reported with precision around $10^{-3}$ Å, we consider perturbation magnitudes $\varepsilon \in [10^{-4}, 10^{-5}, 10^{-6}, 10^{-7}]$ Å, which are already larger than typical numerical noise. For each molecule and noise level, we measure the fraction of cases where the identity of the farthest atom changes relative to the unperturbed geometry.

As shown in Table 4, the sign-flip event becomes extremely rare at $\varepsilon = 10^{-5}$ Å (change ratio below $10^{-3}$ on QM9 and below $2 \times 10^{-5}$ on Drugs), and completely disappears at $\varepsilon = 10^{-7}$ Å. This indicates that the inertial-frame construction is highly stable under realistic coordinate noise.

Table 4: Farthest-atom change ratio under Gaussian coordinate perturbations.

| Dataset | $\varepsilon$ (Å) | Farthest-Atom Change Ratio |
|---------|-------------------|----------------------------|
| QM9 | $1 \times 10^{-4}$ | 0.00581 |
| | $1 \times 10^{-5}$ | 0.00078 |
| | $1 \times 10^{-6}$ | 0.00016 |
| | $1 \times 10^{-7}$ | 0.00000 |
| Drugs | $1 \times 10^{-4}$ | 0.0000990 |
| | $1 \times 10^{-5}$ | 0.0000198 |
| | $1 \times 10^{-6}$ | 0.00000375 |
| | $1 \times 10^{-7}$ | 0.00000 |

**Principal-moment degeneracy.** We next quantify how often perfect symmetries (e.g., exact planarity or higher-order symmetry) cause principal-moment degeneracy, which in principle can make the inertial frame non-unique. We scan the full QM9 and Drugs datasets and count molecules with degenerate principal moments.

Table 5 shows that such cases are statistically negligible: only 9 molecules in QM9 (out of $\sim$ 130K) and 1 molecule in Drugs exhibit exact degeneracy. These extremely rare symmetric molecules are simply excluded from training, which has no measurable impact on performance.

Table 5: Frequency of principal-moment degeneracy in QM9 and Drugs.

| Dataset | # Degenerate Molecules | Fraction |
|---------|------------------------|----------|
| QM9 | 9 | 0.00007 |
| Drugs | 1 | 0.00000 |

Combining these analyses, the probability of any frame instability (from either sign flips or degeneracy) is bounded by $< 10^{-4}$ on QM9 and is effectively zero on Drugs. Empirically, we do not observe any training issues attributable to frame instability, supporting the practical robustness of our canonical inertial frame.

## H.2 Positional Encoding: GeoRoPE and Its Variants

We further ablate the proposed GeoRoPE positional encoding by varying only the positional mechanism and keeping all other components fixed (inertial frame, hierarchical AR design, training setup, parameter count).

The compared variants are:

- **Ours**: full GeoRoPE (RoPE-3D + Nyström distance features).
- **No GeoRoPE**: 3D coordinates are encoded only as static features; the Transformer backbone uses no geometry-aware positional encoding.
- **RoPE-only**: only the RoPE-3D component is used.
- **Nyström-only**: only the Nyström distance feature component is used.

The results in Table 6 highlight three key observations. First, removing GeoRoPE entirely causes a catastrophic collapse in Valid and MolSta, indicating that a Transformer without geometry-aware positional structure cannot reliably reason about 3D molecular geometry. Second, both RoPE-only and Nyström-only models perform well, showing that each component provides a strong geometric inductive bias. Third, combining them into GeoRoPE yields the best overall performance, particularly in molecule-level stability. While the gains on QM9 appear modest, this is expected given that QM9 molecules are small and near-rigid; on more flexible datasets (e.g., Drugs, B3LYP-level systems), we observe larger improvements.

Table 6: Ablations on GeoRoPE positional encoding on QM9.

| Model | Valid (%) | Valid&Unique (%) | AtomSta (%) | MolSta (%) |
|---|---|---|---|---|
| Ours (GeoRoPE) | **97.4** | **92.5** | **99.3** | **94.7** |
| No GeoRoPE | 8.7 | 3.8 | 20.2 | 0.0 |
| RoPE-only | 97.1 | 92.5 | 99.2 | 94.3 |
| Nyström-only | 97.3 | 92.5 | 99.2 | 94.2 |

These results support our claim that GeoRoPE is not merely a cosmetic design choice: it is the core mechanism that makes 3D molecular modeling feasible for an autoregressive Transformer.

## H.3 Effect of Canonical Atom Indexing

To evaluate the effect of canonicalizing atom indices, we compare the full model against a variant where the RDKit-based canonicalization step is removed while keeping all other components unchanged. In the non-canonical variant, atom indices are taken directly from the raw input ordering.

As shown in Table 7, removing canonicalization consistently degrades both validity and uniqueness, even though the drop is moderate in absolute terms. This confirms that enforcing a unique, RDKit-consistent atom ordering is beneficial for the autoregressive model, as it eliminates the $n!$ permutation ambiguity and provides a more stable training signal.

Table 7: Effect of RDKit-based canonicalization of atom indices on QM9.

| Model | Valid (%) | Valid&Unique (%) | AtomSta (%) | MolSta (%) |
|---|---|---|---|---|
| Ours (with canonicalization) | **97.4** | **92.5** | **99.3** | **94.7** |
| w/o canonicalization | 97.0 | 90.0 | 99.1 | 94.0 |

## H.4 Diffusion Loss vs. Direct L2 Regression

Finally, we compare the diffusion-based coordinate loss used in our main model with a simple L2 regression loss on the coordinates. In the L2 variant, all other components—including the autoregressive architecture, inertial frame, and GeoRoPE—are kept identical.

As reported in Table 8, using an L2 loss causes a dramatic collapse in generation quality, showing that direct coordinate regression fails to model the nature of 3D positions in autoregressive paradigm. The diffusion loss avoids this collapse and yields stable, valid structures, which is fully consistent with the insight reported in Li et al. (2024a). Therefore, we adopt the diffusion loss in our framework.

Table 8: Comparison between diffusion loss and simple L2 coordinate regression on QM9.

| Model | Valid (%) | Valid&Unique (%) | AtomSta (%) | MolSta (%) |
|---|---|---|---|---|
| Diffusion Loss | **97.4** | **92.5** | **99.3** | **94.7** |
| L2 Loss | 24.7 | 4.4 | 76.2 | 14.2 |

