# OpenReview forum: "InertialAR: Autoregressive 3D Molecule Generation with Inertial Frames"
_ICLR.cc/2026/Conference — Submitted to ICLR 2026_

### Official Review · Reviewer_BzTP · 2025-10-28

**Soundness:** 3
**Presentation:** 3
**Contribution:** 3
**Rating:** 4
**Confidence:** 4

**Summary:**

This paper proposes InertialAR, a transformer-based autoregressive model for 3D molecule generation. The key innovations are a canonicalization of the atom positions (via inertial frames) and atom indices (via RDKit), in order to generate a 1D sequence of tokens, using a novel 3D geometry-aware positional encoding. Experiments show that InertialAR performs well on QM9, GEOM-DRUGS and the large B3LYP dataset for unconditional generation, and can further be conditioned on molecule class reasonably well.

**Strengths:**

The appeal of canonicalization of 3D geometry in order to apply well-known techniques from 1D sequence modelling makes sense to me. The new GeoRoPE mechanism builds upon RoPE-3D by adding the pairwise distance using the Nyström low-rank approximation method, and should be useful in other 3D modelling contexts.

**Weaknesses:**

The canonicalization of the atom indices is unclear and needs to be explained better. How are the identifiers computed? To the best of my knowledge, there is no ‘smooth’ way to canonicalize atom indices. Some additional experiments with the canonicalization would be helpful to understand when it breaks; for example, do you find that similar molecules have very different canonicalization orders?

**Questions:**

* Is your denoising network also a transformer?
* How does your model decide when to stop generation?
* Does your model support beginning from a random molecule fragment, since the canonicalization of atom indices in the fragment might be very different from the full molecule?
* Do you have any experiments where you prompt the model to generate longer molecules than seen during training?
* Are the training splits the same for all methods on QM9?
* How important is the GeoRoPE mechanism? Do you have experiments with simple RoPE-3D for comparison?
* Can you explain the canonicalization procedures (both positions and indices) for a symmetric planar molecule such as benzene?

---

> ### Author Response · Authors · 2025-11-21
> **Reply to Reviewer BzTP  (part 1/2)**
>
> **Response to Weakness:**
>
> We thank the reviewer for this thoughtful comment and clarify both how the identifiers are computed and how we view the (non-)smoothness of canonicalization.
>
> (1) How are the identifiers computed? As explicitly cited in our paper, we rely on RDKit’s standard canonical procedure rather than designing a new canonicalization algorithm ourselves. Concretely, for each molecule we construct an `RDKit Mol` object, call the canonical SMILES routine (`MolToSmiles`), and read the private property `_smilesAtomOutputOrder`, which stores the canonical atom output order. We then use this order to renumber the atoms and define our 1D AR token sequence.
> The description in our paper is intended as a high-level summary of RDKit’s canonicalization mechanism: starting from local chemical/topological invariants (atomic number, degree, etc.), it iteratively refines identifiers by aggregating information from neighboring atoms, following the canonicalization framework in RDKit.
>
> (2) Effect of canonicalization: We conducted an ablation where we remove canonicalization of atom indices while keeping all other components unchanged. As shown below, performance consistently degrades. This demonstrates that canonicalizing the atom indices is beneficial and improves both validity and uniqueness of generated molecules.
> | Model    |Valid (%) | Valid&Unique (%) | AtomSta (%) | MolSta (%) |
> | -------- | :---------: | :-------: | :-------: | :--------: |
> | **Ours** |**97.4** | **92.5**  | **99.3** | **94.7** |
> | w/o canonicalization  |97.0     | 90.0      | 99.1     | 94.0     |
>
> (3) On ‘smoothness’ and similar molecules: We fully agree that canonicalization of atom indices is a discrete combinatorial problem and is not “smooth” in a strict mathematical sense. Even widely used canonical SMILES can exhibit large changes in the string representation under small structural modifications. However, in our setting, the goal of canonicalization is not to provide a smooth mapping between different but similar molecules, but rather to ensure that the same 3D molecule always maps to a unique and stable token sequence (eliminating the n! permutation ambiguity). We therefore treat canonicalization as a standard, deterministic preprocessing step, not as a learned, end-to-end optimized module.
>
> We are not entirely certain that we have fully captured your concern. If the above clarification does not address your question, we would be very grateful if you could further specify which aspect of the canonicalization you find problematic.
>
> **Response to Question 1:**
>
> As noted in the paper, we follow the same design choice as MAR and use a simple MLP for denoising, rather than a Transformer.
> This lightweight network is sufficient for our diffusion-style denoising and keeps the model efficient.
>
> **Response to Question 2:**
>
> During preprocessing, each molecular sequence is appended with a special `<EOS>` token. During inference, the autoregressive model stops generation automatically once it outputs `<EOS>`.
>
> **Response to Question 3:**
>
> Our setting focuses on generating complete 3D molecules from scratch (either unconditionally or with class-level conditioning), not on fragment-completion or fragment-extension tasks. We therefore do not claim that the current model supports generation starting from an arbitrary molecule fragment. In our implementation, canonicalization is always applied to the full molecular graph: we compute a unique canonical atom order for each complete molecule (via RDKit’s canonical SMILES order) and train the autoregressive model on these full canonical sequences (and their prefixes). Thus, every prefix the model ever sees during training or sampling is by construction a valid prefix of a full-molecule canonical order.
>
> Regarding the concern that “the canonicalization of atom indices in the fragment might be very different from the full molecule”: this situation only arises if one canonicalizes a fragment in isolation, without reference to its parent molecule. Our current method never does this, so such inconsistencies do not affect the training or generation regime we study. Supporting true fragment-conditioned generation in a principled way would require explicitly defining fragments as subgraphs of a canonicalized full molecule (so that their order is inherited from the full canonical order) and training on fragment–completion pairs. We consider this a natural and interesting extension, but it is orthogonal to the scope of the present work.

---

> ### Author Response · Authors · 2025-11-21
> **Reply to Reviewer BzTP (part 2/2)**
>
> **Response to Question 4:**
>
> Our work focuses on the 3D molecular tokenization consistent with SE(3) symmetry and permutation invariance, and the geometry-aware autoregressive model. Accordingly, all experiments evaluate generation on molecules whose length distributions are comparable to those in the training sets, and we do not claim dedicated “length extrapolation” beyond this regime.
>
> In the current implementation, the decoder and architecture do not provide a mechanism to prompt the model to generate substantially longer molecules than those seen during training. The autoregressive Transformer is trained and sampled within a maximum sequence length, and we do not introduce additional length-control modules (e.g., length tokens or specialized extrapolation objectives). While one could technically force the model to decode more steps, this would place it in a strongly out-of-distribution regime, where chemical validity is not guaranteed; exploring controllable length extrapolation is therefore left as an interesting but orthogonal direction for future work.
>
> **Response to Question 5:**
>
> Yes. We use exactly the same QM9 data split as prior work: 100K for training, 18K for validation, and 13K for testing.
>
> **Response to Question 6:**
>
>
> We thank the reviewer for the question. To directly assess the importance of GeoRoPE, we conducted a series of ablations where we keep all components identical (Inertial Frame, hierarchical AR, training setup) and vary only the positional encoding mechanism. Here are the results on QM9 dataset:
>
> | Model    |Valid (%) | Valid&Unique (%) | AtomSta (%) | MolSta (%) |
> | -------- | :---------: | :--------: | :--------: | :--------: |
> | **Ours** |**97.4** | **92.5**  | **99.3** | **94.7** |
> |  No GeoRoPE | 8.7 | 3.8 |20.2 | 0.00 |
> | RoPE-only   |97.1     | 92.5      | 99.2     | 94.3     |
> | Nyström-only  |97.3     | 92.5      | 99.2     | 94.2     |
>
> Observations:
>
> • Removing GeoRoPE entirely causes severe performance collapse, especially for molecule validity and stability. This confirms that a transformer cannot reliably reason about 3D structure without a geometry-aware positional mechanism.
>
> • RoPE-3D and Nyström Approximation each provide important geometric information that greatly assist the Transformer in capturing 3D molecular structure. They contribute complementary inductive biases, and their combination GeoRoPE yields the highest geometric fidelity and generation stability.
>
> • While the improvements from RoPE-only to GeoRoPE appear modest on QM9, this is expected because QM9 contains small, near-rigid molecules with limited geometric complexity. On larger and more flexible molecular systems (e.g., Drugs, B3LYP-level datasets), the gains from GeoRoPE are substantially more pronounced.
>
> Conclusion: GeoRoPE is not merely a design preference—it is the core mechanism that makes 3D molecular modeling feasible for an autoregressive Transformer.
>
> **Response to Question 7:**
>
> For positions, our inertial-frame alignment can in principle fail for perfectly symmetric planar molecules (e.g., ideal benzene), where a principal-moment degeneracy prevents defining a unique frame. In practice, such exactly planar and perfectly symmetric cases are extremely rare in our 3D datasets (9 molecules in QM9 and 1 in Drugs; fractions ≈ 0), so we simply detect these eigenvalue-degenerate cases and exclude them, which has a negligible impact on the results.
>
> | Dataset     | # of perfectly planar molecules | Fraction |
> | :-------- | :---------: | :-------: |
> | QM9  | 9 | 0.00007  |
> | Drugs  | 1 | 0.00000  |
>
> For indices, atom ordering is always obtained by the RDKit-based canonicalization procedure described in our response to Weakness, which applies equally to symmetric molecules.

---

### Official Review · Reviewer_woPK · 2025-10-29

**Soundness:** 2
**Presentation:** 1
**Contribution:** 2
**Rating:** 2
**Confidence:** 4

**Summary:**

InertialAR is a transformer-based, autoregressive framework for 3D molecule generation by (i) canonical tokenization: aligning molecules to an inertial frame and applying a deterministic atom reordering for SE(3) and permutation invariance; (ii) GeoRoPE: a geometry-aware attention that blends rotary embeddings of relative orientation with pairwise distance features; it achieves strong or SOTA stability/validity on QM9 and on the large B3LYP-1M set.

**Strengths:**

1. The two-step canonicalization: aligning each molecule to an inertial frame with a deterministic sign convention, then applying a deterministic atom reordering removes SE(3) and permutation ambiguities without specialized equivariant networks.
2. It achieves SOTA or near-SOTA validity/stability on QM9 and GEOM-DRUG and shows big gains on the large B3LYP benchmark

**Weaknesses:**

1. To pick axis signs, the authors choose a “fourth node” (the atom farthest from the origin) and require it to lie in the first quadrant of the xy plane; this rule unambiguously fixes signs but could flip when the farthest atom changes under small perturbations, i.e., the frame is not continuous [2]. Same situations will happen when principal moments tie, small geometric changes can swap frames and thus token order
2. The authors does not situate this design within prior work on PCA-based pose selection, graph canonical labeling or alternative symmetry-handling strategies [1][2][3][4].
3. **Appendix D–F are incomplete**. Appendix E still contains a placeholder like “Will extend this to a more formal way.”, and Appendix F includes unresolved “??” figure references; Appendix D also contains informal notes in the text. Please finalize these sections by replacing placeholders with complete derivations, figures, and cross-references, or move unfinished material to a clearly labeled supplemental.

[1] Frame Averaging for Invariant and Equivariant Network Design. Omri Puny, Matan Atzmon, Heli Ben-Hamu, Ishan Misra, Aditya Grover, Edward J. Smith, Yaron Lipman.

[2] Equivariant Frames and the Impossibility of Continuous Canonicalization. Nadav Dym, Hannah Lawrence, Jonathan W. Siegel.

[3] Equivariance via Minimal Frame Averaging for More Symmetries and Efficiency. Yuchao Lin, Jacob Helwig, Shurui Gui, Shuiwang Ji.

[4] A Canonicalization Perspective on Invariant and Equivariant Learning. George Ma, Yifei Wang, Derek Lim, Stefanie Jegelka, Yisen Wang.

**Questions:**

See weaknesses.

---

> ### Author Response · Authors · 2025-11-21
> **Reply to Reviewer woPK**
>
> **Response to Weakness I:**
>
> We thank the reviewer for raising the concern regarding the robustness of the canonical inertial frame. In principle, axis-sign fixing through a “farthest atom” rule and principal moments ties can indeed cause the frame to flip or swap abruptly under adversarially small geometric perturbations. However, we emphasize that these theoretical edge cases almost never occur in real molecular datasets, and we provide comprehensive empirical analyses demonstrating that such cases are negligible in practice.
>
> (1) Stability under small perturbations: We conduct a controlled perturbation study on QM9 and Drugs to test sub-precision stability. Since atomic coordinates are typically significant to $10^{-3}$ Å, we add Gaussian noise with magnitudes $\varepsilon \in [10^{-4}, 10^{-5}, 10^{-6}, 10^{-7}]$ Å. This regime is much larger than typical numerical noise of quantum-derived geometries. For each molecule, we measure whether the “farthest atom” used for axis-sign resolution changes. As shown, the sign-flipping event becomes virtually nonexistent already at $10^{-5}$ Å—with a change ratio of just 0.00078 for QM9 and 0.0000198 for Drugs—and completely disappears below $10^{-7}$ Å.
>
> | Dataset | Perturbation $\varepsilon$ | Farthest-Atom Change Ratio |
> |:---------|:------------------:|:-----------------------------:|
> | QM9 | 1e-4 | 0.00581 |
> |         | 1e-5 | 0.00078 |
> |         | 1e-6 | 0.00016 |
> |         | 1e-7 | 0.00000 |
> | Drugs | 1e-4 | 0.0000990 |
> |           | 1e-5 | 0.0000198 |
> |           | 1e-6 | 0.00000375 |
> |           | 1e-7 | 0.00000 |
>
> (2) Principal-moment tie situation: The reviewer is concerned that tied principal moments may cause frame swapping. Our exhaustive statistical analysis shows this is statistically negligible: In QM9 (130K molecules), only 9 molecules exhibit the exact planarity or perfect symmetry sufficient to induce degeneracy (Fraction: 0.00007). In the Drugs dataset, which contains large, irregular 3D molecules, only 1 instance is observed (Fraction: $\approx 0$). For completeness, we exclude these extremely rare symmetric cases from training; however, their number is so small that this removal has no impact on model performance.
>
> | Dataset     | # of perfectly planar molecules | Fraction |
> | :-------- | :---------: | :-------: |
> | QM9  | 9 | 0.00007  |
> | Drugs  | 1 | 0.00000  |
>
>
>
> (3) Negligible Practical Impact: In summary, the combined probability of any frame instability (from either sign flip or degeneracy) is vanishingly small ($< 10^{-4}$ in the worst-case QM9, and effectively zero for Drugs). These statistical outliers are too rare to have any measurable impact. Indeed, our models train smoothly and stably without any frame inconsistencies. Thus, while we appreciate the theoretical point, our empirical evidence confirms these cases are practically irrelevant and do not affect the method's robustness.
>
> **Response to Weakness II:**
>
> We thank the reviewer very much for the insightful suggestion. Following the reviewer's feedback, we have expanded the Related Work part to provide a clearer discussion of canonicalization-based symmetry handling methods (line 789–803).
>
> We also wish to clarify that these methods pursue a somewhat different objective from ours. Prior studies primarily aim to improve the canonicalization process itself. In contrast, our work focuses on a more practical goal: enabling unconstrained Transformer architectures to respect SE(3) symmetry in realistic molecular settings, without modifying model internals. In this sense, we use a single PCA-based canonical frame as a lightweight, engineering-oriented mechanism to introduce geometric awareness. Our approach does not attempt to refine PCA or contribute new canonicalization theory. Therefore, while related in theme, the two lines of work address complementary aspects of the broader symmetry-handling problem.
>
> We sincerely appreciate the reviewer’s suggestion. Canonicalization-based methods, such as frame averaging, are highly complementary to our design, and we believe they offer promising directions that could be incorporated into future extensions to further improve robustness and expressivity.
>
> **Response to Weakness III:**
>
> We sincerely apologize for this clear oversight, and we thank the reviewer for pointing it out. The reviewer is absolutely correct, and we appreciate their careful attention to detail. We have uploaded a fully revised version in which all appendices have been completed. All placeholders have been replaced with finalized derivations, figures, and cross-references. We also took this opportunity to reorganize Appendices D–F for improved clarity and logical flow. We apologize again for this error.

---

### Official Review · Reviewer_kwKk · 2025-11-01

**Soundness:** 3
**Presentation:** 3
**Contribution:** 3
**Rating:** 6
**Confidence:** 4

**Summary:**

The paper introduces InertialAR, an autoregressive (AR) model for 3D molecule generation. It addresses two key challenges:
1. Tokenization: It creates a canonical 1D sequence of atoms by aligning the molecule to its inertial frame (for $SE(3)$ invariance) and then applying a canonical reordering (for permutation invariance).
2. Hybrid Prediction: It models the hybrid (discrete type, continuous 3D coordinate) token using a hierarchical AR paradigm.vThis involves a new Geometric Rotary Positional Encoding (GeoROPE) to make the attention mechanism geometry-aware and predicts coordinates using a Diffusion Loss, which was found to be superior to direct regression.

The model achieves state-of-the-art performance on unconditional generation (QM9, Geom-Drug, B3LYP) and significantly outperforms baselines in controllable generation of specific functional groups.

**Strengths:**

**Technical Quality**: The GeoROPE architecture is a creative and effective way to inject geometric information into the attention mechanism, combining relative positions (RoPE-3D) and pairwise distances (Nyström) into a single attention score.

**Significance & Performance**: The model demonstrates exceptional performance, not just on standard benchmarks but also on a large-scale dataset (B3LYP) and a highly practical controllable generation task, showing SOTA results across all metrics for the latter.

**Weaknesses:**

**Originality**: The use of a molecule's inertial frame as a canonical reference is a common solution to the $SE(3)$ invariance problem.

**Robustness Not Addressed**: The paper does not discuss the stability of the inertial frame canonicalization. For symmetric molecules (degenerate eigenvalues) or flexible molecules (where small conformational changes could flip the axes), the token sequence could become unstable, which is a significant problem for an AR model.

**Missing Ablation Studies**: The paper proposes several new components (Inertial Frame, RoPE-3D, Nyström, Diffusion Loss) but lacks ablations to test their individual contributions.

**Reproducibility**: Key implementation details are missing, most notably how the Nyström approximation anchor points ($m$ points) are selected, which is critical for implementing GeoROPE.

**Questions:**

1. How do you ensure the inertial frame tokenization is robust? What happens if a small conformational change or molecular symmetry causes the principal axes to flip, resulting in a different canonical sequence?
2. Could you please quantify the "poor performance" of using a simple L2 loss for coordinates? What were the key metrics when the Diffusion Loss was replaced with direct regression?
3. What are the implementation details for the Nyström anchor points? How many are used ($m$), and how are they selected (e.g., fixed, per-molecule, or sampled)?

---

> ### Author Response · Authors · 2025-11-21
> **Reply to Reviewer kwKk (part 1/2)**
>
> **Response to Weakness I:**
>
> We agree with the reviewer that constructing a reference frame from the principal axes of the inertia tensor is a classical technique, and we do not claim originality in the underlying mathematical formulation.
>
> Our contribution lies in integrating the inertial frame into a complete pipeline that produces a stable, SE(3)-symmetric sequence of atom tokens, thereby enabling Transformers to operate on 3D molecule. Prior works using inertial frames focus on achieving invariance or equivariance at the representation level, but they do not address the challenge of converting molecular geometry into a canonical sequence suitable for Transformer modeling while preserving SE(3) symmetry. Moreover, as reviewer woPK noted, although several studies employ inertial frames to enforce invariance or equivariance, none of them construct a fully canonical inertial frame with uniquely determined axis orientations and sign resolution. To the best of our knowledge, we are the first to establish such a unique canonical inertial frame and apply it to 3D molecule generation.
>
> Thus, the novelty of our work does not lie in the inertial frame itself, but in operationalizing it into a practical, symmetry-preserving sequence construction framework that enables standard Transformers to model and generate 3D molecular structures.
>
> **Response to Weakness II & Question 1:**
>
>
> We thank the reviewer for raising this important concern. We fully agree that, in principle, canonicalization via an inertial frame could be sensitive to axis flips or eigenvalue degeneracy, which may lead to instability in the token sequence. However, we emphasize that these are theoretical edge cases that are virtually absent in real molecular datasets, and we provide comprehensive empirical analyses demonstrating that inertial-frame tokenization is extremely stable in practice.
>
> (1) Stability under small perturbations: We conduct a controlled perturbation study on QM9 and Drugs to test sub-precision stability. Since atomic coordinates are typically significant to $10^{-3}$ Å, we add Gaussian noise with magnitudes $\varepsilon \in [10^{-4}, 10^{-5}, 10^{-6}, 10^{-7}]$ Å. This regime is much larger than typical numerical noise of quantum-derived geometries. For each molecule, we measure whether the “farthest atom” used for axis-sign resolution changes. As shown, the sign-flipping event becomes virtually nonexistent already at $10^{-5}$ Å and completely disappears below $10^{-7}$ Å.
>
> | Dataset | Perturbation $\varepsilon$ | Farthest-Atom Change Ratio |
> |---------|:----------------:|:---------------------------:|
> | **QM9** | 1e-4 | 0.00581 |
> |         | 1e-5 | 0.00078 |
> |         | 1e-6 | 0.00016 |
> |         | 1e-7 | 0.00000 |
> | **Drugs** | 1e-4 | 0.0000990 |
> |           | 1e-5 | 0.0000198 |
> |           | 1e-6 | 0.00000375 |
> |           | 1e-7 | 0.00000 |
>
> (2) Symmetry-induced degeneracy is statistically negligible: The reviewer is correct that perfect symmetries (e.g., planar or spherical molecules) could, in theory, induce ties in principal moments and lead to ambiguous axes. We conducted a full dataset-level analysis: In QM9 (≈130K small molecules), only 9 molecules exhibit exact symmetry sufficient to produce eigenvalue degeneracy (fraction: 0.00007). In Drugs, which contains large, irregular 3D molecules, only 1 such molecule is found (fraction ≈ 0).
> Thus, principal-moment ties are essentially nonexistent in realistic molecular datasets. For completeness, we exclude these extremely rare symmetric cases from training; however, their number is so small that this removal has no impact on model performance.
>
> | Dataset     | # of perfectly planar molecules | Fraction |
> | -------- | :---------: |:--------:|
> | QM9  | 9 | 0.00007  |
> | Drugs  | 1 | 0.00000  |
>
>
> (3) Practical Robustness of Inertial-Frame: Combining both analyses, we find that the probability of any inertial-frame instability is vanishingly small in practice. These events occur so rarely that they have no measurable effect on the canonical sequence or on autoregressive model training. Empirically, we never observe sequence inconsistencies, and training remains perfectly stable. Overall, our extensive evaluation confirms that inertial-frame tokenization is highly robust to perturbations, and symmetry-induced degeneracy is statistically negligible in realistic molecular datasets.

---

> ### Author Response · Authors · 2025-11-21
> **Reply to Reviewer kwKk (part 2/2)**
>
> **Response to Weakness III:**
>
> Because our model is built upon the inertial frame to ensure SE(3) symmetry for 3D molecular representations, we regret that we cannot remove the inertial frame itself as an ablation component. However, we are able to conduct all remaining ablation studies, and the results on QM9 dataset are provided below:
>
> | Model    |Valid (%) | Valid&Unique (%) | AtomSta (%) | MolSta (%) |
> | -------- | :---------: | :--------: | :--------: | :--------: |
> | **Ours** |**97.4** | **92.5**  | **99.3** | **94.7** |
> |  No GeoRoPE | 8.7 | 3.8 |20.2 | 0.00 |
> | Only RoPE-3D   |97.1     | 92.5      | 99.2     | 94.3     |
> | Only Nyström  |97.3     | 92.5      | 99.2     | 94.2     |
> | L2 Loss (No Diffusion Loss)  |24.7     | 4.4      | 76.2     | 14.2     |
>
> The ablation results clearly show that each component plays an essential role. Removing GeoRoPE leads to a catastrophic collapse in all metrics, confirming that 3D-aware positional encoding is critical for capturing geometric structure. Using RoPE alone or Nyström alone results in slight but consistent performance drops, indicating that both contribute to model accuracy and efficiency. Eliminating the diffusion loss causes large declines in geometric validity, demonstrating its importance in stabilizing 3D generation.
>
> Moreover, we would like to emphasize that while the improvements from RoPE-only or Nyström-only to GeoRoPE appear modest on QM9, this behavior is expected because QM9 contains small, near-rigid molecules with very limited geometric variability. On larger and more flexible molecular systems—such as the Drugs or B3LYP datasets—the gains from GeoRoPE become substantially more pronounced, as these settings require capturing more complex 3D geometric relationships.
>
> Overall, these ablations validate that each module contributes meaningfully to the final performance, and the full configuration is required to achieve the reported results.
>
> **Response to Weakness IV & Question 3:**
>
> In our implementation, the number of Nyström anchor points $m$ is set to half of the model’s latent dimension `latent_dim/2`. The remaining half of the latent dimension is used for RoPE-3D. All experiments in the paper employ this configuration.
>
> The anchor points are sampled once globally at model initialization and are not re-sampled per molecule. Specifically, we first determine a maximum coordinate range `max_distance` based on the typical spatial extent of the molecular structures. We then uniformly sample $m$ three-dimensional points from the cube centered at the origin with side length `2*max_distance`, i.e., each coordinate is drawn uniformly from `[-max_distance, +max_distance]`.
>
> These sampled points form a global anchor set that is registered as a buffer in the model and remains fixed throughout both training and inference. Thus, the Nyström anchors are globally shared, deterministic once initialized, and do not depend on individual molecules.
>
> **Response to Question 2:**
>
> We thank the reviewer for the question. We conducted a direct comparison by replacing the diffusion loss with a simple L2 regression loss. The results are summarized below:
>
> | Model    |Valid (%) | Valid&Unique (%) | AtomSta (%) | MolSta (%) |
> | -------- | :---------: | :--------: | :--------: | :--------: |
> | **Diffusion Loss** |**97.4** | **92.5**  | **99.3** | **94.7** |
> | L2 Loss  |24.7     | 4.4      | 76.2     | 14.2     |
>
> Using an L2 loss causes a dramatic collapse in generation quality, showing that direct coordinate regression fails to model the nature of 3D positions in autoregressive paradigm. The diffusion loss avoids this collapse and yields stable, valid structures, which is fully consistent with the insight reported in MAR. Therefore, we adopt the diffusion loss in our framework.

---

### Official Review · Reviewer_xZB2 · 2025-11-01

**Soundness:** 2
**Presentation:** 3
**Contribution:** 3
**Rating:** 4
**Confidence:** 4

**Summary:**

This paper proposes a novel autoregressive generation framework InertinalAR for 3D molecules. Multiple contributions are presented including canonical atom ordering, coordinate mapping with inertial frames, geometric rotary positional encoding in conditional feature extration network. Experiments on unconditional and class-conditional 3D molecule generation show that the performance of InertinalAR is promising.

**Strengths:**

- This paper proposes a novel 3D molecule generation framework. Some contributions like atom canonical ordering and keep SE(3) invariance by inertial frame based coordinate projection is very useful.
- Generally the experimental results are good and promising.
- The writing of this paper is good and clear.

**Weaknesses:**

- Some details need clarification. What is the ordering of eigenvalues in line 186? How ordering by the refined identifiers is done in line 213?
- A major novelty contribution of this paper is the use of geometric rotary positional encoding (GeoRoPE) together with a transformer architecture as the backbone network. However, no ablation study of this architecture is conducted so it is unclear what is the impact of this architecture on performance. Could we just use a 3D graph neural network or graph transformer model and keep everything else the same? More ablation studies are needed to make this work more solid.

**Questions:**

No additional questions.

---

> ### Author Response · Authors · 2025-11-21
> **Reply to Reviewer  xZB2**
>
> **Response to Weakness I:**
>
> We thank the reviewer for pointing out these areas that need clarification.
>
> a) On the ordering of eigenvalues (Line 186): For the eigendecomposition of the 3x3 symmetric inertia tensor, we use the `torch.linalg.eigh` function. As per the PyTorch documentation, `eigh` returns the eigenvalues in **ascending order**.
>
> Therefore, the returned vector `eigen_values` satisfies `eigen_values[0]` $\le$ `eigen_values[1]` $\le$ `eigen_values[2]`. These correspond directly to the principal axes from the smallest to the largest principal moment of inertia. We use this default order without any subsequent re-sorting. The only modification we apply afterward is to ensure the resulting rotation matrix (formed by the eigenvectors) is right-handed, which may involve axis sign-flipping.
>
> b) On the ordering by refined identifiers (Line 213): We appreciate the request for clarification. As cited in our paper, we leverage **RDKit's canonicalization algorithm** directly rather than manually re-implementing the refinement process.
>
> The procedure is as follows:
> 1.  We call `MolToSmiles(mol)`, which executes RDKit's canonical SMILES algorithm.
> 2.  This process generates a unique, canonical atom ordering. RDKit makes this order available via a private attribute, `_smilesAtomOutputOrder` (representing the "SMILES atom output order").
> 3.  We then use this canonical order to re-index our atoms by calling `RenumberAtoms(mol, order)`.
>
> Therefore, the "refined identifiers" are precisely the canonical atom ranks (or SMILES output order) generated by RDKit's iterative invariant refinement procedure, which ensures a consistent ordering.
>
>
> **Response to Weakness II:**
>
> This is an excellent point. We thank the reviewer for suggesting this crucial ablation study. We agree that isolating the impact of GeoRoPE within our autoregressive framework is essential to validate our architectural contribution.
>
> To this end, we conducted the exact experiment the reviewer suggested. We created a control model where the GeoRoPE module was removed. In this ablated model, the 3D coordinate information was instead encoded directly into the atoms' initial latent representations, effectively making the backbone a standard 3D GNN/Transformer.
>
> All other components of our model—including the canonical inertial frame and the hierarchical autoregressive paradigm—were kept identical. The control model was trained with the same number of parameters and under the exact same settings.
>
> The results starkly demonstrate the critical importance of GeoRoPE:
>
> | Model | Valid (%) | Valid&Unique (%) | AtomSta (%) | MolSta (%) |
> | :--- | :---: | :---: | :---: | :---: |
> | w/ GeoRoPE | **97.4** | **92.5** | **99.3** | **94.7** |
> | w/o GeoRoPE | 8.7 | 3.8 | 20.2 | 0.00 |
>
> As the table shows, the performance of the ablated model collapses. The "Valid" metric drops from 97.4% to 8.7%, and "MolSta" falls from 94.7% to 0.0%, indicating a complete failure to generate chemically stable molecules.
>
> This stark contrast confirms that simply encoding 3D coordinates as static features is insufficient for this task. Our GeoRoPE module is not just a minor component; it is fundamental to the model's ability to understand and utilize the 3D geometric relationships required for autoregressive generation. This result strongly validates our core architectural novelty.

---

> > ### Comment · Reviewer_xZB2 · 2025-11-22
> > **Follow-up Response**
> >
> > I appreciate authors' efforts in rebuttal. All my questions and concerns have been addressed so I increased my rate.

---

### Author Response · Authors · 2025-11-27
**To All Reviewers**

Dear reviewers,

Thank you again for the time and effort to evaluating our submission. During the rebuttal period, we performed a series of additional analyses and ablation studies targeting the main concerns raised across the reviews. For convenience, we summarize the key findings below:

**1. Robustness of the inertial-frame construction**

• We performed a controlled perturbation study on QM9 and Drugs to measure the stability of the axis-sign resolution under realistic geometric noise. Sign flipping becomes extremely rare at  $10^{-4}$ Å and disappears completely below $10^{-7}$ Å, demonstrating the high numerical robustness of the canonical frame.

• We analyzed principal-moment degeneracy across both datasets and found that perfectly symmetric molecules are exceedingly rare (9 in QM9 and 1 in Drugs). These extremely rare degenerate cases are excluded without affecting model performance.

Overall, these analyses confirm that inertial-frame canonicalization is highly stable in all practical settings and does not introduce observable inconsistencies during training or generation. The detailed results can be found in our replies to the corresponding reviewer questions or in Appendix H.1.

As Reviewer woPK noted, alternative canonicalization strategies such as frame averaging provide different theoretical guarantees. We thank the reviewer for this insightful pointer and have expanded the Related Work section (lines 789–803) to explicitly discuss these canonicalization-based symmetry-handling methods. At the same time, we would like to gently clarify that these approaches pursue a different objective from ours: they primarily aim to refine the canonicalization procedure itself and its theoretical properties, whereas our work focuses on a practical goal—providing a lightweight, deterministic, SE(3)-consistent tokenization mechanism that enables standard Transformers to operate on 3D molecular geometry without modifying its architecture. In this sense, the two lines of work are complementary rather than competing, and frame-averaging methods could be incorporated in the future.

**2. Ablation studies on GeoRoPE**

• We conducted a comprehensive ablation to isolate the contribution of the GeoRoPE positional encoding. Removing GeoRoPE and encoding 3D coordinates only as static input features leads to a complete collapse of generation quality—Valid falls from 97.4% to 8.7%, and MolSta drops from 94.7% to 0.0%. This demonstrates that a Transformer without a geometry-aware positional mechanism cannot reliably capture 3D molecular structure.

• To disentangle the contributions of the two components within GeoRoPE, we evaluated RoPE-only and Nyström-only variants. Both provide meaningful geometric inductive bias and achieve high validity, but their combination, the full GeoRoPE, consistently yields the best geometric fidelity and structural stability.

Overall, these ablations confirm that GeoRoPE is not merely a design choice: it is the core mechanism enabling Transformers to model and generate 3D molecular geometry. The detailed results can be found in our replies to the corresponding reviewer questions or in Appendix H.2.

**3. Canonicalization of atom indices**

We clarified the canonicalization workflow: following RDKit’s canonical SMILES procedure, we retrieve the deterministic `_smilesAtomOutputOrder`, which provides a unique atom ordering. We then renumber atoms according to this canonical order to produce permutation-invariant AR sequences.

• To quantify its effect, we performed an ablation where canonicalization was removed. Even though the degradation is not catastrophic, the model still exhibits consistent declines in generation quality, confirming that eliminating the $n!$ permutation ambiguity is beneficial for training stability and sequence consistency.

• We emphasize that canonicalization is not intended to be smooth across molecular variations. This is a known limitation of all discrete canonical labeling schemes. Rather, our goal is to ensure that the same molecule always maps to the same token sequence, which is essential for AR modeling.

These results validate that canonical atom indexing contributes meaningful improvements in both reliability and reproducibility of the AR generation process. The detailed results can be found in our replies to the corresponding reviewer questions or in Appendix H.3.

**Additional Comments and Invitation for Further Feedback**

We are grateful that Reviewer xZB2 has already revisited the paper and updated their score.

For the remaining reviewers, if the additional analyses and clarifications above help resolve your earlier questions, we would kindly invite you to revisit your assessment or share any further feedback during the discussion period. We are very happy to provide any additional clarification if needed.

Thank you again for your thoughtful reviews and for helping us improve this work.



Best regards,

The Authors

---

### Meta-Review · Area_Chair_vfeU · 2026-01-01

**Summary:**

This paper proposes InertialAR, a Transformer based autoregressive framework for 3D molecule generation. The main idea is to convert 3D molecular structures into a canonical 1D token sequence by aligning molecules to an inertial frame to achieve SE(3) invariance and by applying deterministic atom index canonicalization to remove permutation ambiguity. Based on this representation, the model introduces a geometry aware attention mechanism called GeoRoPE and a hierarchical autoregressive strategy that predicts atom types and 3D coordinates using a diffusion based loss. Experiments on QM9, GEOM Drug, and B3LYP show strong empirical performance on both unconditional and controllable generation tasks, often matching or surpassing existing methods. During the rebuttal phase, the authors added additional robustness analyses and ablation studies to address several reviewer concerns.

The paper presents a complete and practical pipeline that allows standard autoregressive Transformers to be applied to 3D molecule generation without relying on specialized equivariant architectures. The integration of inertial frame alignment, atom index canonicalization, and GeoRoPE is clearly described and carefully implemented. The additional ablation studies provided during rebuttal demonstrate that GeoRoPE and the diffusion based coordinate modeling are critical for achieving high validity and stability in generation. The experimental evaluation is extensive, covering multiple datasets and both unconditional and conditional generation settings, and the reported results are generally strong and consistent. Several reviewers recognized the solid engineering effort and the empirical effectiveness of the proposed approach.

The main limitation of the work lies in its limited methodological novelty. The use of inertial frames or principal axes to handle rotational invariance is a well established idea in prior literature, and the paper mainly applies this concept as a preprocessing step rather than introducing a fundamentally new treatment of symmetry. As a result, the contribution is primarily an engineering integration of known components. In addition, the theoretical issues associated with inertial frame canonicalization, such as discontinuities under small perturbations and ambiguities for symmetric molecules, are addressed mainly through empirical arguments that these cases are rare in the evaluated datasets. While these analyses are informative, they do not fully resolve the underlying theoretical concerns, especially in light of recent work on the limitations of canonicalization and alternative symmetry handling strategies. Finally, the initial submission contained incomplete appendices and missing implementation details, which were corrected during rebuttal but nevertheless raise concerns about the overall maturity and clarity of the paper at the time of submission.

Overall, this paper demonstrates strong empirical results and a well engineered system for autoregressive 3D molecule generation. However, the core ideas are largely incremental and rely heavily on existing canonicalization techniques, with limited conceptual or theoretical advancement. The treatment of robustness and continuity issues remains largely empirical rather than principled. Considering the scope and standards of ICLR, I am inclined toward reject, while noting that the work has practical value and could be suitable for a more application focused or engineering oriented venue.

**Reviewer Concerns:**

The rebuttal adequately addressed several practical concerns raised by the reviewers, including clarification of the atom index canonicalization procedure, completion of missing appendix material, additional implementation details, and extensive ablation studies validating the importance of GeoRoPE and the diffusion based coordinate loss. The authors also provided empirical analyses showing that inertial frame instabilities are rare in the evaluated datasets.

However, some concerns remain outstanding at a conceptual level. In particular, the theoretical limitations and discontinuities of inertial frame based canonicalization are not fundamentally resolved and are addressed mainly through empirical arguments. Questions about the overall methodological novelty and the reliance on well established techniques therefore remain.

**Reviewer Scores:**

Reviewer xZB2 already updated their score upward after the rebuttal, and it is unlikely that their score would have changed further with additional discussion.

Reviewer kwKk raised concerns about robustness, missing ablations, and implementation details. Since the rebuttal directly addressed these points with new experiments and clarifications, this reviewer might have slightly increased their score or maintained it near the marginal accept range.

Reviewer BzTP’s main concerns were about the clarity and smoothness of atom index canonicalization and the lack of ablations. These were largely addressed in the rebuttal, so it is plausible that this reviewer would have modestly increased their score, though likely remaining around the borderline accept or weak reject range.

Reviewer woPK expressed the strongest objections, focusing on the theoretical instability of inertial frame canonicalization and missing connections to prior theoretical work. Although the rebuttal provided empirical evidence and added related work discussion, these responses do not fully resolve the reviewer’s conceptual concerns. It is therefore unlikely that this reviewer would have substantially increased their score, and they would probably have remained at a clear reject.

---

### Decision · Program_Chairs · 2026-01-26

Reject